# Self-sampling monkeypox virus testing in high-risk populations, asymptomatic or with unrecognized Mpox, in Spain

Cristina Agustí [1,2,3] ✉, Héctor Martínez-Riveros [1,3,4],
Àgueda Hernández-Rodríguez[5,6], Cristina Casañ[5], Yesika Díaz [1],
Lucía Alonso [1,7], Elisa Martró [2,3,5,6], Jordana Muñoz-Basagoiti [8,14],
Marçal Gallemí [8,14], Cinta Folch [1,2,3], Ibrahim Sönmez [1,3], Héctor Adell[9],
Marta Villar[9], Alexia París de León [5], Sandra Martinez-Puchol [3,5,10],
A. C. Pelegrin [3,5], Daniel Perez-Zsolt [8], Dàlia Raïch-Regué [8], Rubén Mora [9],
Luis Villegas[9], Bonaventura Clotet[3,7,8,11,12], Nuria Izquierdo-Useros [3,8,12],
Pere-Joan Cardona[5,6,13] & Jordi Casabona[1,2,3,6]

The recent monkeypox virus (MPXV) outbreak was of global concern and has mainly affected gay, bisexual and other men who have sex with men (GBMSM). Here we assess prevalence of MPXV in high-risk populations of GBMSM, trans women (TW) and non-binary people without symptoms or with unrecognized monkeypox (Mpox) symptoms, using a self-sampling strategy. Anal and pharyngeal swabs are tested by MPXV real-time PCR and positive samples are tested for cytopathic effect (CPE) in cell culture. 113 individuals participated in the study, 89 (78.76%) were cis men, 17 (15.04%) were TW. The median age was 35.0 years (IQR: 30.0–43.0), 96 (85.02%) individuals were gay or bisexual and 72 (63.72%) were migrants. Seven participants were MPXV positive (6.19% (95% CI: 1.75%–10.64%)). Five tested positive in pharyngeal swabs, one in anal swab and one in both. Six did not present symptoms recognized as MPXV infection. Three samples were positive for CPE, and showed anti-vaccinia pAb staining by FACS and confocal microscopy. This suggests that unrecognized Mpox cases can shed infectious virus. Restricting testing to individuals reporting Mpox symptoms may not be sufficient to contain outbreaks.

Mpox is a zoonotic disease caused by monkeypox virus (MPXV), a virus belonging to the Orthopoxvirus genus, which is endemic in several African countries[1]. From 1 January through 12 December 2022, a cumulative total of 82,628 laboratory-confirmed cases of Mpox and 65 deaths were reported to the World Health Organization (WHO) from 110 countries[2]. On 23 July 2022 the WHO declared Mpox to be a Public Health Emergency of International Concern[3]. Spain, with 7412 cases, has been the third-most affected country after the United States of America and Brazil[2]. In Spain, as in other countries, the outbreak has mainly affected gay, bisexual and other men who have sex with men

(GBMSM) with no documented history of travel to countries where MPXV is endemic.

MPXV infection can cause genital, perianal, oral lesions as well as complications like proctitis and tonsillitis[4]. Some authors suggest that, instead of respiratory transmission, the dominant transmissibility mode of MPXV in non-endemic Mpox countries is local inoculation by close skin-to-skin contact during sexual activity[4]. However, sexual transmission by means of semen has not been ruled out[5].

Diagnosis of MPXV infection is based on nucleic acid amplification testing, using quantitative or qualitative polymerase chain reaction

(PCR). The recommended specimen type for laboratory confirmation of MPXV is skin lesion material including; swabs of lesion surface and/or exudate, roofs from more than one lesion, or lesion crusts[6]. MPXV testing is recommended for suspected cases presenting with symptoms that suggest this type of infection. However, two previous studies have reported positive MPXV PCR results among asymptomatic individuals[7,8]. This supports the hypothesis that a proportion of MPXV infections remain undiagnosed, either because individuals have no symptoms (asymptomatic/pre-symptomatic infections), or because their symptoms are not attributed to a possible MPXV infection (unrecognized infections)[8]. Furthermore, very few studies have explored whether these infections could contribute to viral transmission between individuals.

Ward et al. [9] performed a contact tracing study, linking data on case-contact pairs of MPXV infection on probable exposure dates in the UK between 6 May and 1 August 2022. They demonstrated that more than half (53%) of the transmission events in the UK outbreak occurred in the pre-symptomatic phase of infection. Furthermore, they estimated that transmission occurred up to four days before the onset of symptoms[9]. Retrospective PCR detection in patients of sexual health clinics in France and Belgium suggests that some patients could have an asymptomatic MPXV infection[7,8]. In symptomatic Mpox cases, a study performed in a cohort of patients with relatively mild disease showed MPXV viral clearance within the first 2 months following the appearance of symptoms and a shorter duration of the period when replication-competent virus was detected in viral cultures[10].

The transmission dynamics of MPXV in the current outbreak are highly consistent with a sexually transmitted infection (STI)[11]. Although a few women and children have been infected since May 2022, the majority of cases in Spain have occurred among GBMSM. The 2022 MPXV outbreak shows some similarities with the HIV epidemic regarding potential stigmatization of key populations, such as GBMSM[12]. Stigma can prevent access to care for diagnosis and, in turn, prevent contact tracing and other containment measures. Alternative STI testing modalities such as self-testing and self-sampling constitute important options to diversify and optimize testing access and studies have demonstrated that they increase uptake of STI testing for all groups, including those at high-risk[13-15]. Furthermore, a recent study showed that the performance of diagnostic tests from self-collected samples was similar to that of physician-collected samples, suggesting that self-sampling is a reliable strategy for diagnosing MPXV infection[16].

In the present study, we aimed (i) to assess the prevalence of MPXV infection among highly exposed GBMSM and trans women (TW), asymptomatic or with mild unrecognized Mpox symptoms, who were recruited in a community-based centre in Barcelona, (ii) to assess the presence of replication-competent particles of MPXV and (iii) to evaluate the feasibility and acceptability of a community-based self-sampling strategy for Mpox diagnosis.

## Results
### Characteristics of participants
From August to October 2022, 113 individuals were recruited at a community centre in Barcelona and participated in the study. The main characteristics of the participants are shown in Table 1. From all the participants, 89 (78.76%) were cis men, 17 (15.04%) were TW and 3 (2.65%) non-binary gender. Additionally, 4 participants chose not to disclose their gender or did not provide a clear response. The median age of participants was 35.0 years (Interquartile Range (IQR): 30.0–43.0), 96 (85.02%) individuals were gay or bisexual and 72 (63.72%) were migrants; mainly from Colombia (29.17%), Venezuela (16.67%), Brazil (11.11%), Italy (5.56%) and Argentina (5.56%). Additionally, 44 (38.94%) participants self-reported HIV infection and among HIV negative participants 41 (59.42%) were on PrEP and 58 (51.33%) had had an STI in the previous 12 months. Regarding MPXV, 28 (24.78%)

participants had had contact with a confirmed Mpox case over the previous 30 days. Also seven (6.19%) and 13 (11.50%) participants had received the smallpox vaccine in their childhood or in the previous 12 months, respectively. In addition, 80 (70.80%) individuals were extremely or moderately concerned about Mpox and 53 (46.90%) considered it likely or very likely that they would get an MPXV infection (Table 1).

Behavioural characteristics of participants are shown in Table 2. The median number of sexual partners of participants over the previous 30 days was 5.00 (IQR: 1.00–10.00), of the total participants 42 (39.25%) had not used condoms during sexual intercourse over the previous month and 38 (33.63%) had had sex in exchange for money, gifts or favours. Furthermore, 29 (30.85%) had practiced chemsex in the previous 30 days and three (5.45%) had practiced slamming in the last month. Significant differences between participants with a positive or negative result for MPXV infection were found for the following variables: having practiced double penetration (vagina and anus) ($P = 0.004$), slamming ($p = 0.041$), having met their sexual partners in music festivals in the last month ($p = 0.027$) and taking their shirt off while partying ($p = 0.033$) (Table 2).

### MPXV prevalence and characteristics of individuals testing positive for MPXV
Analyses of 113 pharyngeal and 112 anal swabs, respectively, were performed in the reference laboratory. Eight positive MPXV results for seven individuals were detected and we estimated a total prevalence of 6.19% (95% CI: 1.75–10.64%). All positive participants were cis gay men and prevalence in this group was 7.87% (95% CI: 2.27–13.46%).

The characteristics of the participants with a positive MPXV result are shown in Table 3. Five participants tested positive in pharyngeal swabs, one in the anal swab and one in the pharyngeal and the anal swabs. PCR cycle thresholds (Ct) ranged from 24.85 to 38.06; and the viral load from 2674 to 8,532,000 copies/mL.

We aimed to isolate viable virus from the pharyngeal and anal swab samples with reported PCR positive results for MPXV. We included two undetectable samples along with media as negative controls. As a positive control, we used a viral stock previously isolated during the 2022 summer belonging to the same outbreak and geographical location. We cultured the samples and followed them up for 14 days or until cytopathic effect (CPE) was detected under the microscope when infected cells were assessed for the detection of viral antigens using an anti-vaccinia polyclonal antibody (pAb) by fluorescence-activated cell sorting (FACS) and confocal microscopy. Although two viral cultures had to be discarded due to the presence of contaminant microorganisms, we were able to follow up ten for 14 days (Table 3). Three out of the six PCR positive samples successfully cultured over time (40, 66 and 95) were positive, not only for CPE at the indicated day of harvesting, but also for staining for specific anti-vaccinia pAb detected by FACS and confocal microscopy. Sample 72 was at the limit of positivity (TCID$_{50}$ < 10) and was not considered as a positive (Table 3 and Supplementary Fig. 1). The Ct values of samples 40, 66 and 95 ranged from 24.85 to 36.79, indicating that samples with low viral load, such as the one collected by individual 95, were competent for cellular infection in vitro (Table 3). We also tested the recovered viral stocks by PCR, which yielded positive results for MPXV, and we sequenced them along with the positive control viral stock, which confirmed the specificity of the MPXV presence in cell cultures, while we detected no signal in the negative control. Furthermore, infectivity was measured as tissue culture infectious doses per ml (TCID$_{50}$/mL; Table 3). Negative samples and media were negative for all of the analyses, while the inoculation with a viral stock resulted in positive CPE and viral antigen detection by FACS and confocal microscopy (Table 3 and Supplementary Fig. 1). These results demonstrated a propagation of infectious MPXV from at least three

**Table 1 | Main characteristics of participants from Stop Mpox**

| | Mpox virus infection status | | |
|---|---|---|---|
| | All N = 113 | Negative N = 106 | Positive N = 7 |
| **Median age (IQR)** | 35.00 [30.00;43.00] | 35.00 [30.00;43.00] | 37.00 [34.00;48.50] |
| **Gender** | | | |
| Cis man | 89 (78.76%) | 82 (77.36%) | 7 (100.00%) |
| Trans woman | 17 (15.04%) | 17 (16.04%) | 0 (0.00%) |
| Non binary person | 3 (2.65%) | 3 (2.83%) | 0 (0.00%) |
| DK/DA | 4 (3.54%) | 4 (3.77%) | 0 (0.00%) |
| **Sexual orientation** | | | |
| Gay | 84 (74.34%) | 77 (72.64%) | 7 (100.00%) |
| Heterosexual | 8 (7.08%) | 8 (7.55%) | 0 (0.00%) |
| Bisexual | 12 (10.62%) | 12 (11.32%) | 0 (0.00%) |
| Other | 3 (2.65%) | 3 (2.83%) | 0 (0.00%) |
| DK/DA | 6 (5.31%) | 6 (5.66%) | 0 (0.00%) |
| **Country of birth** | | | |
| Spain | 37 (32.74%) | 34 (32.08%) | 3 (42.86%) |
| Other | 72 (63.72%) | 68 (64.15%) | 4 (57.14%) |
| DK/DA | 4 (3.54%) | 4 (3.77%) | 0 (0.00%) |
| **HIV positive** | | | |
| Yes | 44 (38.94%) | 41 (38.68%) | 3 (42.86%) |
| No | 61 (53.98%) | 58 (54.72%) | 3 (42.86%) |
| I don't know my HIV status | 1 (0.88%) | 0 (0.00%) | 1 (14.29%) |
| I do not want to answer | 1 (0.88%) | 1 (0.94%) | 0 (0.00%) |
| DK/DA | 6 (5.31%) | 6 (5.66%) | 0 (0.00%) |
| **Take PrEP regularly** | | | |
| Yes | 41 (59.42%) | 38 (58.46%) | 3 (75.00%) |
| No | 20 (28.99%) | 20 (30.77%) | 0 (0.00%) |
| DK/DA | 8 (11.59%) | 7 (10.77%) | 1 (25.00%) |
| **STI history in the last 12 months** | | | |
| None | 52 (46.02%) | 49 (46.23%) | 3 (42.86%) |
| Syphilis | 20 (17.70%) | 20 (18.87%) | 0 (0.00%) |
| Chlamydia | 20 (17.70%) | 19 (17.92%) | 1 (14.29%) |
| Gonorrhea | 24 (21.24%) | 22 (20.75%) | 2 (28.57%) |
| Lymphogranuloma venereum | 1 (0.88%) | 1 (0.94%) | 0 (0.00%) |
| Genital herpes | 3 (2.65%) | 3 (2.83%) | 0 (0.00%) |
| Human papillomavirus | 5 (4.42%) | 5 (4.72%) | 0 (0.00%) |
| *Mycoplasma genitalium* | 7 (6.19%) | 7 (6.60%) | 0 (0.00%) |
| Hepatitis A | 0 (0.00%) | 0 (0.00%) | 0 (0.00%) |
| Hepatitis B | 0 (0.00%) | 0 (0.00%) | 0 (0.00%) |
| Hepatitis C | 3 (2.65%) | 2 (1.89%) | 1 (14.29%) |
| Other STIs | 6 (5.31%) | 6 (5.66%) | 0 (0.00%) |
| DK/DA | 3 (2.65%) | 1 (0.94%) | 2 (28.57%) |
| **Contact with a confirmed case of Mpox** | | | |
| Yes | 28 (24.78%) | 25 (23.58%) | 3 (42.86%) |
| No | 63 (55.75%) | 60 (56.60%) | 3 (42.86%) |
| DK/DA | 22 (19.47%) | 21 (19.81%) | 1 (14.29%) |
| **Traveled in the last 30 days** | | | |
| Yes | 53 (46.90%) | 50 (47.17%) | 3 (42.86%) |
| No | 57 (50.44%) | 53 (50.00%) | 4 (57.14%) |
| DK/DA | 3 (2.65%) | 3 (2.83%) | 0 (0.00%) |

**Table 1 (continued) | Main characteristics of participants from Stop Mpox**

| | Mpox virus infection status | | |
|---|---|---|---|
| | All N = 113 | Negative N = 106 | Positive N = 7 |
| **Having received the Smallpox vaccine** | | | |
| Yes, vaccinated in childhood | 7 (6.19%) | 6 (5.66%) | 1 (14.29%) |
| Yes, vaccinated in the last 12 months | 13 (11.50%) | 12 (11.32%) | 1 (14.29%) |
| No | 75 (66.37%) | 71 (66.98%) | 4 (57.14%) |
| DK/DA | 18 (15.93%) | 17 (16.04%) | 1 (14.29%) |
| **Mpox Concern Level** | | | |
| Extremely concerned | 40 (35.40%) | 38 (35.85%) | 2 (28.57%) |
| Moderately concerned | 40 (35.40%) | 37 (34.91%) | 3 (42.86%) |
| Somewhat concerned | 17 (15.04%) | 15 (14.15%) | 2 (28.57%) |
| Slightly concerned | 6 (5.31%) | 6 (5.66%) | 0 (0.00%) |
| Not at all concerned | 1 (0.88%) | 1 (0.94%) | 0 (0.00%) |
| DK/DA | 9 (7.96%) | 9 (8.49%) | 0 (0.00%) |
| **Self-preceived probability of getting Mpox** | | | |
| Very likely | 18 (15.93%) | 17 (16.04%) | 1 (14.29%) |
| Likely | 35 (30.97%) | 31 (29.25%) | 4 (57.14%) |
| Neither very nor unlikely | 29 (25.66%) | 27 (25.47%) | 2 (28.57%) |
| Unlikely | 13 (11.50%) | 13 (12.26%) | 0 (0.00%) |
| Very unlikely | 4 (3.54%) | 4 (3.77%) | 0 (0.00%) |
| DK/DA | 14 (12.39%) | 14 (13.21%) | 0 (0.00%) |

*IQR* Interquartile range, *DK/DA* don't know/don't answer.
August - October 2022. Barcelona (Spain). *N*: 113.

out of the six (3/6) samples that were successfully cultured over time from the original 8 MPXV positive samples identified.

The PCR test produced an inconclusive result for eight anal samples due to the absence of human DNA (myostatin gene). This lack of detection could have been caused by the presence of inhibitors or an incorrect sample collection process by the user.

Regarding presentation of symptoms, none of the participants reported any symptoms at the time of recruitment. After three weeks, participants with a positive result of MPXV were contacted by phone to enquire if they had had any symptom before testing them and within the following 21 days, there was no information available regarding one of the participants with a positive MPXV result (1/7), two (2/7) positive-testing participants reported having no symptoms before testing, or 21 days after. One (1/7) had no symptoms before testing and reported having fever, exhaustion, sore throat and a skin lesion in the 21 days following testing positive. Three (3/7) participants reported the following symptoms before testing: a swollen inguinal lymph node, fever, exhaustion and a skin lesion and none of those participants connected these symptoms with MPXV infection. (Table 3). Taking into account only participants without symptoms before testing (excluding symptomatic participants), the estimated prevalence of MPXV infection was 2.65% (95 CI%: 0–5.62). However, it was decided not to exclude participants who presented mild unrecognized symptoms from the study due to the potential relevance of those individuals in transmission of the infection. It is worth noting that viable MPXV viruses were obtained only from individuals who reported symptoms. Two of these individuals reported experiencing mild symptoms prior to testing, although they did not relate them to Mpox, while one individual did not display

## Table 2 | Behavioural characteristics of participants from Stop Mpox

| | Mpox virus infection status | | |
|---|---|---|---|
| | All N = 113 | Negative N = 106 | Positive N = 7 |
| **Number of sexual partners last 30 days (Median, IQR)** | 5.00 [1.00;10.00] | 5.00 [1.00;10.00] | 3.00 [1.00;5.00] |
| **Sexual practices last 30 days** | | | |
| Mutual masturbation | 96 (89.72%) | 90 (90.00%) | 6 (85.71%) |
| Insertive oral sex | 99 (92.52%) | 92 (92.00%) | 7 (100.00%) |
| Receptive oral sex | 88 (82.24%) | 82 (82.00%) | 6 (85.71%) |
| Insertive back kiss/ rimmimg | 76 (71.03%) | 72 (72.00%) | 4 (57.14%) |
| Receptive back kiss/ rimmimg | 62 (57.94%) | 58 (58.00%) | 4 (57.14%) |
| Insertive vaginal sex | 8 (7.48%) | 8 (8.00%) | 0 (0.00%) |
| Receptive vaginal sex | 4 (3.74%) | 4 (4.00%) | 0 (0.00%) |
| Doble penetration (vagina and anus) | 4 (3.74%) | 2 (2.00%) | 2 (28.57%) |
| Double or triple anal penetration | 10 (9.35%) | 9 (9.00%) | 1 (14.29%) |
| Fist/fisting by anus and/ or vagina | 14 (13.08%) | 12 (12.00%) | 2 (28.57%) |
| Golden shower (urinating on another person) | 24 (22.43%) | 22 (22.00%) | 2 (28.57%) |
| Scat play | 1 (0.93%) | 1 (1.00%) | 0 (0.00%) |
| Using/sharing sex toys | 24 (22.43%) | 22 (22.00%) | 2 (28.57%) |
| Threesome | 47 (43.93%) | 45 (45.00%) | 2 (28.57%) |
| Group sex | 32 (29.91%) | 30 (30.00%) | 2 (28.57%) |
| Other | 2 (1.87%) | 2 (2.00%) | 0 (0.00%) |
| **Condom use sexual intercourse last 30 days** | | | |
| Never | 42 (39.25%) | 37 (37.00%) | 5 (71.43%) |
| Less than half time | 13 (12.15%) | 12 (12.00%) | 1 (14.29%) |
| Around half | 8 (7.48%) | 8 (8.00%) | 0 (0.00%) |
| More than half | 10 (9.35%) | 10 (10.00%) | 0 (0.00%) |
| Always | 31 (28.97%) | 30 (30.00%) | 1 (14.29%) |
| Not applicable (not having had penetrative sex) | 1 (0.93%) | 1 (1.00%) | 0 (0.00%) |
| DK/DA | 2 (1.87%) | 2 (2.00%) | 0 (0.00%) |
| **Do you take off your shirt when you're out partying** | | | |
| Very likely | 25 (22.12%) | 25 (23.58%) | 0 (0.00%) |
| Likely | 11 (9.73%) | 8 (7.55%) | 3 (42.86%) |
| Neither very nor unlikely | 11 (9.73%) | 10 (9.43%) | 1 (14.29%) |
| Unlikely | 20 (17.70%) | 20 (18.87%) | 0 (0.00%) |
| Very unlikely | 36 (31.86%) | 33 (31.13%) | 3 (42.86%) |
| DK/DA | 10 (8.85%) | 10 (9.43%) | 0 (0.00%) |
| **Where did you meet sexual partners in the last 30 days** | | | |
| Associations (LGTBI+ organizations, sports club, etc) | 3 (2.80%) | 3 (3.00%) | 0 (0.00%) |
| Cafe or bar | 8 (7.48%) | 8 (8.00%) | 0 (0.00%) |
| Nightclub | 22 (20.56%) | 22 (22.00%) | 0 (0.00%) |
| Dark room | 8 (7.48%) | 7 (7.00%) | 1 (14.29%) |
| Sex club | 11 (10.28%) | 10 (10.00%) | 1 (14.29%) |
| Sauna | 13 (12.15%) | 10 (10.00%) | 3 (42.86%) |
| Gym | 7 (6.54%) | 5 (5.00%) | 2 (28.57%) |
| Chemsex session | 13 (12.15%) | 11 (11.00%) | 2 (28.57%) |
| Cruising area | 11 (10.28%) | 9 (9.00%) | 2 (28.57%) |
| Gay dating apps/webs | 48 (44.86%) | 45 (45.00%) | 3 (42.86%) |

## Table 2 (continued) | Behavioural characteristics of participants from Stop Mpox

| | Mpox virus infection status | | |
|---|---|---|---|
| | All N = 113 | Negative N = 106 | Positive N = 7 |
| Mainstream dating apps/webs | 21 (19.63%) | 20 (20.00%) | 1 (14.29%) |
| Social media | 13 (12.15%) | 12 (12.00%) | 1 (14.29%) |
| Zoom, Tumblr | 1 (0.93%) | 1 (1.00%) | 0 (0.00%) |
| Music festivals | 10 (9.35%) | 7 (7.00%) | 3 (42.86%) |
| Mass events (pride march, etc) | 4 (3.74%) | 4 (4.00%) | 0 (0.00%) |
| Already known sexual partners | 31 (28.97%) | 29 (29.00%) | 2 (28.57%) |
| Other | 14 (13.08%) | 14 (14.00%) | 0 (0.00%) |
| DK/DA | 4 (3.74%) | 3 (3.00%) | 1 (14.29%) |
| **Sex in exchange for money, gifts or favors (life time)** | | | |
| Yes | 38 (33.63%) | 37 (34.91%) | 1 (14.29%) |
| No | 68 (60.18%) | 62 (58.49%) | 6 (85.71%) |
| DK/DA | 7 (6.19%) | 7 (6.60%) | 0 (0.00%) |
| **Use drugs for sex** | | | |
| Never | 39 (41.49%) | 36 (40.91%) | 3 (50.00%) |
| Yes, in the last month | 29 (30.85%) | 28 (31.82%) | 1 (16.67%) |
| Yes, in the last 6 months | 9 (9.57%) | 7 (7.95%) | 2 (33.33%) |
| Yes, in the last 12 months | 6 (6.38%) | 6 (6.82%) | 0 (0.00%) |
| Yes, more than 12 months ago | 5 (5.32%) | 5 (5.68%) | 0 (0.00%) |
| DK/DA | 6 (6.38%) | 6 (6.82%) | 0 (0.00%) |
| **Have you practiced slam or slamming:** | | | |
| Never | 42 (76.36%) | 41 (78.85%) | 1 (33.33%) |
| Yes, in the last month | 3 (5.45%) | 2 (3.85%) | 1 (33.33%) |
| Yes, in the last 6 months | 1 (1.82%) | 1 (1.92%) | 0 (0.00%) |
| Yes, in the last 12 months | 1 (1.82%) | 0 (0.00%) | 1 (33.33%) |
| Yes, more than 12 months ago | 3 (5.45%) | 3 (5.77%) | 0 (0.00%) |
| DK/DA | 5 (9.09%) | 5 (9.62%) | 0 (0.00%) |

IQR interquartile range, DK/DA don't know/don't answer.
August - October 2022. Barcelona (Spain). N: 113.

any symptoms prior to testing but reported them within the following 21 days. These results highlight that pre-symptomatic phases of infection have the potential to promote ongoing viral transmission events in the community.

### Acceptability and feasibility of the self-sampling intervention

In relation to the acceptability and usability of the self-sampling procedure, 88 (77.87%) and 90 (79.64%) participants considered that the self-sampling procedure was easy or very easy for pharyngeal and anal swabs respectively; and 99 (87.61%) and 98 (86.72%) agreed or agreed very strongly with the statement "I reckon I have collected the pharyngeal sample correctly" and "I reckon I have collected the anal sample correctly", respectively (Table 4). In addition, 98 (86.73%) participants were satisfied or very satisfied with the self-sampling screening intervention and 103 (99.0%) agreed or agreed very strongly that they would recommend participating in the intervention to a friend. The most commonly identified advantages of the intervention by participants were: (i) privacy and confidentiality (75.22%) and (ii) that the test was free (74.34%) (Table 4). The most preferred place to repeat the MPXV test if necessary was a community-based centre (53.10%) and most preferred performing self-sampling at home (48.67%) compared to attending a health care setting (41.59%). (Table 4). These results do

**Table 3 | Results of MPXV infection among participants Stop Mpox**

| Individual | PCR Results | | | | Cell Culture | | | | | | | Symptoms |
|---|---|---|---|---|---|---|---|---|---|---|---|---|
| | Location | Result | Ct | CV (Copies/mL) | Culture viability | Day of Harvesting | CPE | IF | FACS | Ct | TCID$_{50}$/ml | |
| 2 | Pharingeal | Positive | 34.9 | 18,300 | No | 4 | N/A | N/A | N/A | N/A | N/A | No reported symptoms |
| | Anal | Negative | - | - | N/A | N/A | N/A | N/A | N/A | N/A | N/A | |
| 40 | Pharingeal | Positive | 30.1 | 347,000 | Yes | 8 | Positive | Positive | 8.41% | 21.96 | 10^4,3 | No symptoms before testing. After testing the participant reported: Fever, exhaustation, sore throat and a skin lesion |
| | Anal | Negative | - | - | N/A | N/A | N/A | N/A | N/A | N/A | N/A | |
| 64 | Pharingeal | Positive | 37.09 | 4827 | Yes | 8 | Negative | Negative | Negative | Negative | Negative | No available information |
| | Anal | Negative | - | - | N/A | N/A | N/A | N/A | N/A | N/A | N/A | |
| 66 | Pharingeal | Positive | 24.85 | 8,532,000 | Yes | 7 | Positive | Positive | 74.2%* | 20.42 | 10^4,8 | Before testing: A swollen inguinal lymph node |
| | Anal | Positive | 35.35 | 13,960 | No | 2 | N/A | N/A | N/A | N/A | N/A | |
| 72 | Pharingeal | Negative | - | - | N/A | N/A | N/A | N/A | N/A | N/A | N/A | Before testing: Fever, exhaustation and a skin lesion in the genital area |
| | Anal | Positive | 38.06 | 2674 | Yes | 7 | Positive | Positive | 3.75% | 35.98 | <10^1 | |
| 81 | Pharingeal | Positive | 36.99 | 5126 | Yes | 8 | Negative | Negative | Negative | Negative | Negative | No reported symptoms |
| | Anal | Negative | - | - | N/A | N/A | N/A | N/A | N/A | N/A | N/A | |
| 95 | Pharingeal | Positive | 36.79 | 5825 | Yes | 14 | Positive | Positive | 3.18% | 25.88 | 10^2,8 | Skin lesions in the genital area |
| | Anal | Negative | - | - | N/A | N/A | N/A | N/A | N/A | N/A | N/A | |
| 112 | Pharingeal | Negative | - | - | Yes | 8 | Negative | Negative | Negative | Negative | Negative | - |
| 113 | Pharingeal | Negative | - | - | Yes | 8 | Negative | Negative | Negative | Negative | Negative | - |
| Control + | Viral stock | N/A | - | - | Yes | 7 | Positive | Positive | Positive | 19.21 | 10^6,8 | - |
| Control – | Culture media | N/A | - | - | Yes | 8 | Negative | Negative | Negative | Negative | Negative | - |

Samples were inoculated into cell cultures, which were assayed for viability over time (only viable samples with no contamination were followed up), the day of harvesting, the detection of cytopathic effect (CPE), the percentage of infected cells detected by flow cytometry (FACS) and the detection of immunofluorescence against vaccinia antigens (IF). *Sample with asterisk denotes that it had to be sub-cultured again as primary culture exceeded 95% of CPE which precluded proper FACS analysis.
Ct Cycle treshold, CPE Cytopathic Effect, IF Immuno flurencence, FACS Fluorescence Assisted Cell Sorting. Samples assayed to isolate infectious viruses and results obtained.
August - October 2022. Barcelona (Spain).
*Sample with asterisk denotes that it had to be sub-cultured again as primary culture exceeded 95% of CPE wichtpfmt 0precluded proper FACS analysis.

**Table 4 | Usability and acceptability of the self-sampling intervention to detect MPVX**

| | All *N* = 113 |
|---|---|
| **Difficulty level pharyngeal self-sampling** | |
| Very easy | 57 (50.44%) |
| Easy | 31 (27.43%) |
| Neither easy nor difficult | 8 (7.08%) |
| Difficult | 7 (6.19%) |
| Very difficult | 1 (0.88%) |
| DK/DA | 9 (7.96%) |
| **Difficulty level anal self-sampling** | |
| Very easy | 61 (53.98%) |
| Easy | 29 (25.66%) |
| Neither easy or difficult | 9 (7.96%) |
| Difficult | 5 (4.42%) |
| Very difficult | 0 (0.00%) |
| DK/DA | 9 (7.96%) |
| **I trust that I have collected the pharyngeal sample correctly** | |
| Agree strongly | 72 (63.72%) |
| Agree | 27 (23.89%) |
| Neither very nor disagree | 5 (4.42%) |
| Disagree | 0 (0.00%) |
| Disagree strongly | 1 (0.88%) |
| DK/DA | 8 (7.08%) |
| **I trust that I have collected the anal sample correctly** | |
| Agree strongly | 73 (64.60%) |
| Agree | 25 (22.12%) |
| Neither very nor disagree | 5 (4.42%) |
| Disagree | 1 (0.88%) |
| Disagree strongly | 0 (0.00%) |
| DK/DA | 9 (7.96%) |
| **Level of satisfaction self-sampling intervention to detect Mpox** | |
| Very satisfied | 66 (58.41%) |
| Satisfied | 32 (28.32%) |
| Neither satisfied nor unsatisfied | 6 (5.31%) |
| Unsatisfied | 0 (0.00%) |
| Very unsatisfied | 0 (0.00%) |
| DK/DA | 9 (7.96%) |
| **Would you repeat self-sampling to detect Mpox** | |
| Agree strongly | 84 (74.34%) |
| Agree | 16 (14.16%) |
| Neither very nor disagree | 4 (3.54%) |
| Disagree | 0 (0.00%) |
| Disagree strongly | 1 (0.88%) |
| DK/DA | 8 (7.08%) |
| **Would you recommend self-sampling to detect monkeypox to a friend** | |
| Agree strongly | 91 (80.53%) |
| Agree | 12 (10.62%) |
| Neither very nor disagree | 0 (0.00%) |
| Disagree | 0 (0.00%) |
| Very disagree | 1 (0.88%) |
| DK/DA | 9 (7.96%) |
| **Consider self-sampling to detect monkeypox as a good intervention for Mpox screening** | |
| Agree strongly | 86 (76.11%) |
| Agree | 16 (14.16%) |
| Neither very nor disagree | 3 (2.65%) |

**Table 4 (continued) | Usability and acceptability of the self-sampling intervention to detect MPVX**

| | All *N* = 113 |
|---|---|
| Disagree | 0 (0.00%) |
| Disagree strongly | 0 (0.00%) |
| DK/DA | 8 (7.08%) |
| **Self-perceived advantages of self-sampling to detect Mpox** | |
| Privacy and confidentiality | 85 (75.22%) |
| More convenience since you don't have to go to the medical center | 80 (70.80%) |
| The test is free | 84 (74.34%) |
| No prescription needed | 61 (53.98%) |
| No need to give explanations | 76 (67.26%) |
| It contributes to nomalize the monkeypox test | 61 (53.98%) |
| Allows me to take control of my health regarding monkeypox | 73 (64.60%) |
| DK/DA | 2 (1.77%) |
| **Self-perceived disadvantages of self-sampling to detect Mpox** | |
| That the test requires the introduction of a swab orally | 14 (12.39%) |
| That the test requires the introduction of a swab via the anus | 14 (12.39%) |
| Not having the result at the moment | 49 (43.36%) |
| Not having emotional support when receiving the result | 10 (8.85%) |
| The time to receive the result is too long | 16 (14.16%) |
| Other | 5 (4.42%) |
| DK/DA | 22 (19.47%) |
| **Preferred location for repeat testing if it was necessary to repeat the test** | |
| Health care centre | 47 (41.59%) |
| Comunity based centre | 60 (53.10%) |
| Self-sampling at home | 55 (48.67%) |
| DK/DA | 8 (7.08%) |

Stop Mpox. August - October 2022. Barcelona (Spain). *N*: 113.

not only emphasize the acceptance of self-collected samples, but also the feasibility of using this strategy for downstream laboratory analyses involving viral isolation techniques.

## Discussion

Our study provides evidence that there are Mpox cases that remain undiagnosed because patients have no symptoms, or because they have mild unrecognized Mpox symptoms[7,8]. The estimated prevalence of this viral infection was 6.19% for all individuals and 7.87% for cis gay men. Three out of the 7 participants who tested positive for MPXV through our study did not have any symptom before testing (MPXV prevalence taking into account only asymptomatics: 2.65%), while three of them had symptoms that were confused with STIs that cause skin rashes or mucosal lesions, such as genital herpes, syphilis, acuminate condyloma and chancroid among others, as previously described[17]; or even COVID-19 (which causes fever and exhaustion).

Tarin et al.[4] proposed that skin-to-skin contact, rather than the respiratory route, is the dominant mode of MPXV transmission outside endemic countries. This was based on the history of sexual exposure, predominant anogenital skin lesions, and higher viral loads in skin than throat swabs. As, MPXV has been previously isolated from semen[18–20] sexual transmission of MPXV during the 2022 outbreak cannot be ruled out. Our findings show that viable virus can be isolated from pharyngeal swabs.

Our results suggest that restricting testing only to individuals reporting symptoms compatible with MPXV infection may not be enough to contain an ongoing outbreak. In areas with high community

transmission, screening for MPXV in pharyngeal and anal swabs should be offered to those GBMSM at risk of acquiring MPXV.

Although Mpox cases have been reported among TW and non-binary individuals[21], no cases among TW have been detected in our study or previous studies performed in Spain[4,22]. Nevertheless, TW are highly vulnerable to HIV and other STI infections; and in the case of HIV the WHO has recognized the high vulnerability and specific health needs of transgender people with the consequent need for a distinct and independent status in the global HIV response[23]. The disparities in MPXV prevalence between TW and GBMSM could be explained by distinct sexual networks across populations without shared transmission, which is similar to HIV transmission patterns described among TW, their sexual partners and GBMSM[24].

Ubals et al. have recently described that MPXV diagnostic tests with both self-collected swabs and physician-collected swabs have shown a similarly high accuracy and yield similar Ct values in both samples[16]. We demonstrated that a self-sampling intervention for MPXV screening in collaboration with a community centre is feasible and acceptable. It resulted in high levels of satisfaction and willingness to participate from the target population and most of participants considered it easy or very easy to self-collect the samples. Moreover, it also allowed us to perform highly sensitive laboratory techniques downstream, such as viral isolation in cell culture.

Linkage to care is often challenging in self-sampling strategies. We obtained high percentages of confirmation and linkage to care since 6 out of 7 (85.7%) MPXV positive participants self-reported having been linked to care. The linkage to care rate obtained is similar to previous studies on self-sampling strategies for HIV screening also addressed to GBMSM[13,14] and comparable to the percentage of individuals with a reactive screening test for HIV who were linked to care in a network of community-based services, which offer voluntary counselling and testing for HIV in Spain[25]. Follow-up of participants with a positive result should be reinforced to improve rates of linkage to care.

Our study presents several limitations. Firstly, the study population is not representative of GBMSM and TW in Catalonia as we used an opportunistic sample. It is worth noting that the sample had a notably high proportion of migrants and sex workers, likely due to the involvement of a collaborating centre with a specific program for this population. While this may introduce some bias, we did not observe significant differences in the number of sexual partners in the last 30 days (median: 5.00 (IQR: 1.00; 10.00) vs 3.00 (IQR: 1.00;5.00); *p*-value: 0.413) or having history of exchanging sex for money, gifts, or favours (34.91% vs 14.29%; *p*-value: 0.632) between participants with negative and positive MPXV infection results. Secondly, while self-collected anal swabs have been described as a feasible, valid, and acceptable alternative for men who have sex with men and women attending STI clinics[26], an inconclusive PCR result was obtained in 7% in these samples compared to 0% in pharyngeal swabs. Thirdly, we analysed both pharyngeal and anal samples, but for logistical reasons we did not include seminal samples, which, although they may not affect the overall prevalence of MPXV, may be necessary to better define the potential routes of transmission. Fourthly, due to the high sensitivity of PCR techniques, there is the possibility of obtaining false positive results. In this situation, Ct values are high (Ct > 34) and are associated with low viral loads and the laboratory should re-extract and re-amplify the original sample or a new sample requested if this is not possible[27]. In our study, there were three samples with high Cts (64 pharynx; 72 anal; 81 pharynx). Re-testing these samples was not possible due to the low volume left. However, the PCR performed after the viral culture for MPXV was positive for one sample (sample 72). Previous observations have shown that pharyngeal and anal samples exhibit higher Cts in comparison to samples obtained from lesions[4,7]. Furthermore, good practices were followed to prevent contamination and erroneous diagnoses, and negative controls produced the expected result. Moreover, the study's conclusions rely on samples in which the ability

of MPVX to replicate has been verified and validated using diverse and independent methods. Fifthly, it was not feasible to reach the initially expected number of participants, 113 participants instead of 150 were included in the study. Although the precision of the estimates decreased due to a reduction in the expected number of participants, the confidence intervals have remained significant but wider because the obtained prevalence was higher than the prevalence used to calculate the sample size. Finally, the small number of TW tested within this study could have precluded the detection of positive cases in this group.

In conclusion, our findings have important public health implications, particularly for MPXV infection prevention and control policies. We have shown that MPXV infection is present among asymptomatic individuals and among vulnerable populations. We also confirmed that MPXV symptoms can overlap and be confused with other diseases, such as STIs. Moreover, we were able to isolate replication-competent viruses from pharyngeal and anal swabs from asymptomatic or mildly symptomatic patients, as it has been shown in previous studies[8]. First of all, educational interventions are needed to familiarize the members of vulnerable populations with the nature of MPXV symptoms and eradicate the associated stigma in order to increase awareness and health care seeking behaviour in these populations. Secondly, in an epidemic scenario, early diagnosis by means of screening strategies should be aimed not only at suspected clinical cases and direct contacts, but also at all GBMSM at high risk of contracting Mpox, regardless of their symptoms. Community-based self-sampling tools can be acceptable and effective to increase early diagnosis and the eventual isolation of infectious cases. On the other hand, heath care workers in STI clinics, primary care, and emergency rooms in other health care settings should be aware of the variety of Mpox symptoms and the possibility of asymptomatic cases before excluding Mpox as a potential diagnosis. Finally, stigma and discrimination in the most affected group, GBMSM, should be addressed to warrant equitable access to diagnosis, treatments and vaccines. More data are needed to better establish the attributable risk of asymptomatic infections in the transmission of MPXV in an outbreak, including seminal transmission.

## Methods
### Study design and setting
We implemented a transversal non-randomized study offering free self-sampling kits for MPXV testing through a collaborating community centre that offers voluntary counselling and testing for HIV (STOP, Barcelona, Spain). The field coordinator communicated test results to participants by a phone call.

### Study population and recruitment
The study targeted two different key populations: GBMSM and TW. Inclusion criteria were: Self-identification as GBMSM or TW, over 18 years old, with no symptoms of MPXV infection and considered at high risk of contracting Mpox. High risk was defined as: GBMSM and TW who are sex workers and/or chemsex users and/or who practice group sex and/or are HIV positive or are PrEP users. The study was disseminated through Instagram, Facebook, Whatsapp and the community centre website via intermittent campaigns. The campaigns indicated the possibility of get tested for MPXV through a self-sampling intervention free of charge at the headquarters of the collaborating community centre. Participants with eligible criteria were invited to attend the collaborating community centre to get tested for MPXV. No participant was excluded in the study.

The community centre staff, after checking if they comply with inclusion criteria, briefly explained the project to potential participants and obtained the signed informed consent on paper. Participants were provided with comprehensive written information including the study's objectives, the procedures involved, potential benefits of

participation, the voluntary nature of their involvement, and the assurance of confidentiality. Clinical evaluation of symptoms was not conducted during the recruitment process, although, participants were explicitly informed that they could not participate in the study if they presented any symptoms compatible with Mpox. Participants answered a self-completed paper survey on sociodemographic characteristics and behaviour (the survey is available in the Supplementary Information) and self-collected the samples.

It was expected to include 150 participants in the study. A random sample of 142 individuals was established as sufficient to estimate, with 95% confidence and an accuracy of $+/-$ 2 percentage units, a population percentage that was expected to be around 1.5%. The expected prevalence was based in a previous study performed in Belgium among male sexual health clinic attendees with no symptoms of Mpox[8]. After the start of vaccination and the decrease in new cases of Mpox in Spain, the interest of the target population in a screening intervention dropped considerably and it was difficult to reach the expected N (150). Finally, 113 individuals participated in the study.

We collected data prospectively in Barcelona (Spain) from August to October 2022, which was shortly after the peak of the Mpox outbreak in Spain[28].

### Data collection instrument
The following data was collected through the survey: Year and country of birth, sex at birth, gender identity, sexual orientation, level of studies, monthly salary, sex work, having used drugs in the last three months, chemsex in the last three months. Recent Sexual History and risky Practices (<30 days): Number of sexual partners, sexual practices, condom use. Use of PrEP. History of small pox vaccination. Potential MPXV exposures within the last 30 days: Close contact with a Mpox infected case (person I take care of. Sexual contact (groping, masturbation, oral sex, vaginal or anal penetration with or without ejaculation). Shared food, utensils, or dishes. Shared clothes. Towels or bed linen shared at home or elsewhere. Having gone on a trip together. Shared bathrooms (sinks, showers) either at home or elsewhere. Physical contact (face to face, kissing, shaking hands, hugging…). Other); contact with animals, history of travelling, occupational exposure (puncture, laboratory work, contact with potentially contaminated material, healthcare professional without personal protective equipment). History of STI diagnosis. HIV status, PrEP use. Risk perception towards MPXV infection (likelihood of infection, level of concern). Acceptability of the pilot intervention: level of satisfaction, willingness to repeat the experience, likelihood of recommending it to a friend, perceived advantages and disadvantages and preferred way to repeat the MPXV test.

Data related to symptoms on those participants with a positive result was collected through a survey performed through a phone call by the field coordinator after 21 days of receiving test result. The follow-up period was set at 21 days because the incubation period for Mpox is between 5 and 21 days. No clinical assessments were conducted at the time of recruitment.

We used RedCAP version 13.6.1 (REDCap systems, Vanderbilt University, US) to collect data of participants and create an ad hoc online database. We carried out the data entry of the survey data at the coordinating centre.

### Self-sampling kits
The self-sampling kits included an anal and a pharyngeal swab (Molecular Biology Swabs, Deltalab, Rubí, Spain), pre-labelled swab containers and a brochure with detailed instructions with pictures explaining how to get the samples. A video with the instructions of sample collection was available on YouTube and was accessible through a QR code included in the brochure. Participants were able to contact the field coordinator by phone or email if they have any doubt. After obtaining the sample, the swabs were immediately placed in 1 ml

of transport medium (Molecular Biology Swabs, Deltalab, Rubí, Spain) and samples were stored at 4 °C. A parcel courier service provided the secondary and tertiary containers, and samples were transported at 4 °C to the reference laboratory (Microbiology Department, Clinical Laboratory Nord Metropolitan Area, Germans Trias i Pujol University Hospital).

### Delivering test results and follow-up of participants with a positive result
The field coordinator delivered test results by a phone call. All participants with a positive result for MPXV infection were instructed to attend their General Practitioner (GP) or a STI Clinic for follow-up and were advised on isolation measures, sexual abstinence and partner notification. After three weeks, these participants were contacted by phone to enquire if they had had any symptoms before testing them and within the following 21 days.

### Laboratory analysis
**PCR assays.** We analyzed all samples for the detection of MPXV DNA with a real-time PCR-based assay (qualitative and quantitative) at the reference laboratory. We performed nucleic acid extraction using STARMag 96 × 4 Universal Cartridge kit on Seegene StarLet platform (Hamilton Company, Reno, US), according to manufacturer's instructions. We carried out quantitative PCR (qPCR) using the LightMix Modular Monkeypox Virus assay (TIB MolBiol, Berlin, Germany) with LightMix Modular MSTN Extraction Control (TIB MolBiol, Berlin, Germany) as the internal control. We used a thermocycler QuantStudioTM 5 Real-Time PCR System (Applied Biosystems) to amplify an 89 bp-long fragment of the myostatin gene of vertebrates as an internal control and 106 bp-long fragment of the J2L/J2R gene from MPXV. The sequences of the primers used, along with information on the reagents utilized, have been included in the Supplementary Data. We used Applied Biosystems Interpretive Software for detection and data analysis. To determine copy number per mL we used a linear dilution series of a quantified MPXV DNA standard (AMPLIRUN® Monkeypox virus DNA control, Vircell Spain SLU, Santa Fe, Granada, Spain). The calibration curve was composed of 5 points containing 1,000,000, 100,000, 10,000, 1000, and 100 copies/mL (6.00, 5.00, 4.00, 3.00, and 2.00 $\mathrm{Log_{10}}$ copies/mL, respectively), and for each point we analysed three replicates together with negative and positive controls. The MPXV DNA concentration in study samples was extrapolated from the standard curve using the Ct values obtained.

### Cells, viral isolation and titration
We cultured Vero E6 cells (ATCC CRL-1586) at 37 °C and 5% $CO_2$ in Dulbecco's modified Eagle medium (DMEM; Invitrogen) supplemented with 5% foetal bovine serum (Invitrogen), 100 U/mL penicillin and 100 μg/mL streptomycin (all ThermoFisher Scientific, 13179261).

To create the positive control in this study MPXV stock was isolated in August 2022 from a skin lesion swab from a patient diagnosed with Mpox illness. The starting material was the remnant of the specimen participating in Movie study[10], and was irreversibly anonymized to be used as a positive control. Briefly, we cultured Vero E6 cells in T25 culture flasks (25 cm²) at $1.5 \times 10^6$ cells and inoculated them with 1 mL of the liquid sample, for 1 h at 37 °C and 5% $CO_2$. Then we added, 4 ml of 2% FCS-supplemented DMEM containing 100 U/mL penicillin, 100 μg/mL streptomycin and 2,5 μg/mL amphotericin B (all from ThermoFisher Scientific). We maintained cells in incubation and assessed them daily for cytopathic effect (CPE) in order to be able to harvest the supernatant, which was centrifuged at 410 g for 5 min to remove cell debris and stored at −80 °C. We propagated the virus for two passages and collected the supernatant. We titrated the viral stock and confirmed the infection by the presence of viral antigens using antibodies as described below.

## Viral isolation from clinical samples

We inoculated pharyngeal or anal swab samples that had either a positive detection of MPVX DNA by PCR ($n = 8$) or negative results ($n = 2$) into T25 culture flasks with Vero E6 cells as described in the previous section. As a positive control we employed the MPVX stock we had previously isolated, and as a negative control we included mock-treated cells. We assessed viral cultures daily and kept them for 14 days or until 50% CPE was observed. In cases where we detected CPE, we harvested the supernatants, centrifuged them at 410 g for 5 min to remove cell debris and stored them at −80 °C. We washed cells from these positive cultures once with PBS, detached them using 0.5% EDTA trypsin, collected and resuspended them in 0.5 mL of PFA 4% (Merck) for fixation. If we detected no CPE, the cells remained in culture until reaching a confluent state, when we passed half of the cells with supernatant from the previous culture to new flasks and added antibiotics and amphotericin B. We discarded any cultures with undesired microorganism growth and did not consider them for the analysis. We followed the cultures for up to 14 days.

## Immunostaining and FACS analysis

We resuspended fixed cells from positive cultures in 100 μL of Permeabilization Medium (Invitrogen) with a rabbit polyclonal antibody vaccinia virus (Abcam, ab35219) at 1:2000 dilution (2 μg/mL) and incubated them for 20 min at room temperature in darkness. We removed the primary antibody by washing with blocking buffer (PBS, 5% FBS). Then we performed a secondary incubation with a polyclonal goat anti-rabbit IgG H&L Alexa Fluor® 488 antibody (Abcam, ab150077), which we added at 1:1000 dilution (2 μg/mL) and cells were incubated for 20 min at room temperature in darkness. After a PBS wash, we resuspended cells with 300 μL of PBS 1% PFA. To analyze the samples we used a FACSCalibur (Becton-Dickinson) and CellQuest and FlowJo v10.6.1 software to evaluate collected data.

## Confocal microscopy

We prepared microscopy slides (cytospins) with 100 μL of previously fixed and immunostained Vero E6 cells, using EZ Double Cytofunnel (Fisher Scientific) and mounted samples onto slides with Fluoromount-G™ Mounting Medium, with DAPI (Life Technologies). We used a confocal LSM710 microscope and a 63X objective at the IGTP Microscopy Facility to image the cells.

## Viral DNA extraction from viral cultures

We carried out viral DNA extraction with the QiaAmp Viral RNA Mini kit (Qiagen), although this kit has been optimized for the extraction of RNA, it is possible to obtain DNA in parallel according to the manufacturer's instructions. We extracted viral DNA from 140 μL of cellular culture supernatants to perform MPXV PCR as specified.

## Titration of viral isolates from clinical samples

We titrated the viral supernatants collected in 1/10 dilutions using 96-well-plate containing 30.000 Vero E6 cells per well and inspected the plates at the microscope for CPE 6 days post-infection. We were able to calculate the $TCID_{50}$ per mL by inferrence from the number of positive and negative wells using the Reed & Muench method[29].

## MPXV positive control whole genome sequencing

Starting from the DNA extracted from the positive control culture supernatants, we amplified the whole MPXV genome by using the primers and adapting the amplicon tiling approach described by Welkers et al. [30]. Briefly, NextGenPCR® Pre mixed Monkeypox sequencing primer pools (Isogen Lifescience, De Meern, The Netherlands) were used withwith the following cycling conditions: 30 s at 98 °C, and then 35 cycles for 10 s at 98 °C 10 s and 5 min at 65 °C. We prepared sequencing libraries using the Rapid Barcoding Kit 96 from Oxford Nanopore Technologies (ONT, UK) following manufacturer

instructions. Libraries were pooled and loaded onto a R9.4.1 flow cell and sequenced in for 72 h on a MinION Mk1C device (ONT). We processed a negative control along with the samples in order to monitor the whole process. Raw sequencing data (fastq files) were submitted to the European Nucleotide Archive (project PRJEB61105)[31]. Information on used primers and reagents is provided in Supplementary Data 1.

## Bioinformatics analysis of raw sequencing data

Raw sequencing data were analysed using the INSaFLU online platform[32] to obtain consensus genomic sequences. Nextclade (v2.5.0)[33,34] was used to assess consensus quality.

## Statistical analysis

MPXV infection prevalence was estimated by calculating the proportion of individuals with a positive result over the total of individuals with a returned and valid sample. Binomial confidence intervals of 95% were calculated[35].

We carried out descriptive analysis to compare socio-demographic characteristics, risk behaviour variables, and previous STI diagnoses between participants with a positive and negative MPVX test result. Continuous variables were expressed as medians and IQRs. Categorical variables were summarised as absolute values and proportions. For all analysis, a significance level of 5% was considered. All analyses were done using R version 4.0.5.

## Ethical considerations

All identifying data collected was encrypted. Confidentiality was guaranteed in accordance with the provisions of the Regulation (EU) 2016/679 of the European Parliament and of the Council of 27 April 2016 and the new national Organic Law of Protection of Personal Data (3/2018, 5 December, Data Protection and Digital Rights Act). We provided written information about the study to all participants and they had the opportunity to ask questions and clarify queries with the study coordinator by email or phone. The Ethical Committee of the Germans Trias i Pujol Hospital approved the study protocol (PI-22-195). The starting material for the isolation of the control MPXV was provided by the Movie study that was approved by the Ethical Committee of the Germans Trias i Pujol Hospital approved the study protocol (PI-22-156), all participants gave signed an informed consent. The biological biosafety committee of the Germans Trias i Pujol Research Institute approved the execution of MPXV experiments at the BSL3 laboratory of the Centre for Comparative Medicine and Bioimage (CMCiB, protocol number CSB-22-011-M1).

The study was performed in collaboration with a community organization. They participated in the conceptualization of the study, elaboration of the messages, dissemination of the intervention and facilitating access to the target population. Community based centre participation in every phase of the project ensures an approach that takes into account the social, cultural, political context, and views of the target populations.

## Data availability

The raw survey data are protected and are not available due to data privacy laws. The processed survey data are available from the corresponding author upon reasonable request. As mentioned above, raw sequencing data (fastq files) for the positive control have been submitted to the European Nucleotide Archive (project PRJEB61105), it is available at: https://www.ebi.ac.uk/ena/browser/view/PRJEB61105. Source data are provided with this paper.

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

## Acknowledgements

The authors acknowledge the collaboration of Gema Ballega, Pili Bonamusa, Pamela Nef, Ana Isabel Parra Manzano, Enola Crespillo, Aroa Muñoz, Núria Vaquero, Elisa Molina Molina and F. Hoffmann-La Roche Ltd. In addition, the authors thank Harvey Evans for the English revision. This work was supported by the Ministry of Health of Government of Catalonia (Spain) [no grant number] and Hoffmann-La Roche Laboratories. NI-U is supported by the Spanish Ministry of Science and Innovation (grant PID2020-117145RB-I00), EU HORIZON-HLTH-2021-CORONA-01 (grant 101046118) and by institutional funding from Grifols, Pharma Mar, HIPRA, Amassence and Palobiofarma. Finally, the authors thank the CERCA Programme/Generalitat de Catalunya for their support of the Germans Trias i Pujol Research Institute (IGTP).

## Author contributions

Conceptualization: C.A. and J.C.; Funding acquisition: J.C., P.-J.C. and B.C.; Field coordination of the study and follow-up of participants: H.M.-R.; MPXV PCR assays: À.H.-R., C.C., A.P.deL.; MPXV genome amplification, sequencing and bioinformatics analysis: E.M., S.M.-P.,

A.C.P.; Viral extraction, cell culture, titration, immunostaining and FACS analysis: N.I.-U., J.M.-B., M.G., D.P.-Z., D.R.-R.; Statistical analysis: Y.D. and L.A.; Recruitment of participants: H.A., M.V., R.M. and L.V.; Writing-original draft: C.A.; Writing-review & editing: E.M., N.I., J.C., À.H.-R., C.F., I.S., H.M.-R., J.M.-B., M.G., C.C.; All authors contributed to the article and approved the submitted version.

## Competing interests
The authors declare no competing interests. Hoffmann-La Roche Laboratories provided the PCR kits used in the study. The funding body had no role in study design, data collection, data analysis, data interpretation, or writing of the report.

## Additional information

[1]Centre of epidemiological studies on sexually transmitted infections and AIDS of Catalunya (CEEISCAT), Department of Health, Government of Catalonia, Badalona, Spain. [2]Biomedical Research Center Network for Epidemiology and Public Health (CIBERESP), Instituto de Salud Carlos III, Madrid, Spain. [3]Germans Trias i Pujol Research Institute (IGTP), Campus Can Ruti, Badalona, Spain. [4]Doctorate Program in Methodology of Biomedical Research and Public Health, Department of Paediatrics, Obstetrics and Gynaecology and Preventive Medicine, Universitat Autònoma de Barcelona, Badalona, Spain. [5]Microbiology Department, Clinical Laboratory North Metropolitan Area, Germans Trias i Pujol University Hospital, Badalona, Spain. [6]Departament of Genetics and Microbiology, Autonomous University of Barcelona, Badalona, Spain. [7]Fundació Lluita contra las Infeccions, Infectious Diseases Department, Hospital Germans Trias i Pujol, Badalona, Spain. [8]IrsiCaixa AIDS Research Institute, Hospital Germans Trias i Pujol, Badalona, Spain. [9]ONG Stop, Barcelona, Spain. [10]Vicerectorat de Recerca, Universitat de Barcelona, Universitat de Barcelona(UB), Barcelona, Spain. [11]University of Vic–Central University of Catalonia (uVic-UCC), Vic, Spain. [12]Biomedical Research Center Network for Infectious Diseases (CIBERINFEC), Instituto de Salud Carlos III, Madrid, Spain. [13]Biomedical Research Center Network for Respiratory Diseases (CIBERES), Instituto de Salud Carlos III, Madrid, Spain. [14]These authors contributed equally: Jordana Muñoz-Basagoiti and Marçal Gallemí. ✉e-mail: cagusti@iconcologia.net

