## [Peer Review File · Nature Communications]

Self-sampling monkeypox virus testing in high-risk populations, asymptomatic or with unrecognized monkeypox, in SpainReviewers' Comments:

Reviewer #1:

Remarks to the Author:

Line 116 : a "dot" is missing after « migrants »

I have a major concern:

Line 104: in the present study, we aimed (i) to assess the prevalence of MPXV infection among asymptomatic highly exposed GBMSM and trans women (TW) who were recruited in a community-based centre in Barcelona

Line 172: Three (3/6) participants reported the following symptoms before testing: a swollen inguinal lymph node, fever, exhaustion and a skin lesion but none of them connected these symptoms with MPXV infection

These 3 individuals are NOT asymptomatic. Such symptoms can be prodrom symptoms of mpox infection, therefore these individuals should not be counted as "asymptomatic". If authors want to keep them in the analysis, then they should modify their objective and rephrase it : to assess the prevalence of MPXV infection among asymptomatic individuals and in individuals with mild unrecognized Mpox symptoms.

Line 201: Three out of the 7 participants who tested positive for MPXV through our study did not have any symptom before testing, while three of them had symptoms that were easily confused with STIs that cause skin rashes or mucosal lesions, such as genital herpes, syphilis, acuminated condyloma and chancroid among others

This is very confusing. Authors do state that mpox is an STI, or is more likely transmitted via the sexual route, but when symptoms suggestive of STI are reported, authors state that such symptoms are not recognized as possibly related to mpox....

Line 206: We were able to identify replication-competent virus particles in three out of six (3/6) MPXV positive individuals. One of them did not present symptoms before testing and the other two did not recognize their symptoms as indicative of MPXV infection.

◇ again, 2 individuals had STI symptoms. If they did not recognize such symptoms as possibly related to mpox infection, they did seek for medical examination as they did come to the sexual health clinic. So this cannot be considered as a study of prevalence of mpox among "asymptomatic" individuals.

Another concern is on the way the results were given to participants. Were they told to reduce their sexual activity and to notify their partners? The sentence line 210 and 335 are not clear on this matter. Were they followed-up? Was the test redone for confirmation? Line 335 it is mentioned that they were asked to go to their GP or to a sexual health clinic "for confirmation". How many of them had a confirmatory sample? And what was the result of the confirmatory sample?

The discussion section is too long and mixes scientific virological findings with socio-behavioral assumptions and on the acceptability/feasibility on self-sampling, that are not supported by the results presented herein and/or were not part of the objectives of the present study.

Conclusion line 270: We also demonstrate that MPXV symptoms can overlap and be confused with other diseases, such as other STIs. ◇ this has been previously reported and published by other groups such as the US CDC (June – July 2022)

Reviewer #2:

Remarks to the Author:

This is a very interesting paper that focuses on asymptomatic infections of Mpox. It would be interesting to understand possible further transmission although I understand this may not be within the scope of the paper. I have some questions and minor amendments to the manuscript below:

Methods

More details on how the authors arrive at a sample size of 113 participants? How many were excluded and the dropout rate. What biases did this introduce.

Was sample size a consideration in study design?

What biases were present during recruitment?

"participants had received the Mpox vaccine in their childhood or in the previous 12 months, respectively" - do the authors mean the smallpox vaccine in childhood here?

"38 (33.63%) had had sex in exchange for money, gifts or favours" - this suggests a recruitment bias that would be relevant to discuss.

Some high CT values "36.79" that would perhaps warrant a retest per CDC advice "If you obtain a high Ct value (generally ~34 or higher), CDC recommends to immediately re-extract and re-test to ensure there was no cross-contamination."(https://www.cdc.gov/locs/2022/08-23-2022-Lab-Advisory-Monkeypox_Virus_Testing_Considerations_Prevent_False_Positive_Test_Results.html). Worth including this in the limitations.

Line 422 "MPXV infection prevalence was estimated by calculating the proportion of individuals with a positive result over the total of individuals with a returned and valid sample. Confidence interval of 95% was calculated." - This needs expanding. Are they binomial confidence intervals? If so what method?

A suitable alternative method that authors may want to consider for investigating the characteristics of the patients with mpox could be logistic regression.

Reviewer #3:

Remarks to the Author:

The authors performed a study among the clients of a community centre where STI testing was offered. Eligible participants were individuals at high risk for mpox according to a set of criteria. Participants were given a self-sampling kit for anorectal and pharyngeal sampling and surveys about the acceptability of self-sampling were performed concomitantly. The authors found 7 mpox infections among 113 individuals tested, and 6 of them did not report symptoms at the time of testing, nor did 2 of them in the subsequent 21 days. The authors ascertained the presence of viable virus by culture in 4 cases of which 1 did not report any symptoms at the time of testing. The results of this study are in line with the findings of previous studies that found undiagnosed mpox infections, of which the authors cite two.

I'd like to compliment the study team for their achievement to recruit >100 participants among a population that may be difficult to engage in clinical studies, and for their efforts to confirm the results of MPCV culture by several techniques.

Major comments:

1)The main aim of the study was "to assess the prevalence of MPXV infection among asymptomatic highly exposed GBMSM and trans women". Please provide more information in the main text and in the abstract about the targeted population and the way of sampling and data collection so that the reader can obtain an accurate of the population to which the prevalence can be generalised. What was the study design and was this a prospective or retrospective study? When was the study performed

with respect to the mpox epidemic? Who was eligible and who was not? How and where were participants recruited? How and when were their symptoms and signs evaluated? How were clinical and other (survey?) data collected? Two participants reported a swollen lymph node and a skin lesion, so according to the current inclusion criteria, these patients may not have been eligible? Even though it is not clear from the text, I suppose that this information was collected retrospectively, in hindsight? If no thorough clinical assessment was done at the time of recruitment, I would suggest to rephrase the aim of the study and the primary outcome.

2) The statistical analyses in this study are clearly not driven by a prior hypothesis and many of the comparisons are not meaningful, such as the comparison of acceptability of self-testing between test-positive and test-negative cases. Therefore, I recommend to drop all statistical analysis from the manuscript, or only perform those tests that are hypothesis-driven and corrected for multiple testing.

3) Which efforts have been done to exclude that positive results in cases without clear mpox symptoms were caused by contamination? Was infection confirmed on a second sample or through serology?

Minor comments:

4) Introduction, study aim II) "to assess the potential transmissibility of MPXV". I recommend to change wording to avoid the word "transmissibility" because this study was not designed to evaluate MPXV transmission.

5) Abstract: I suggest to not mention viral loads here

6) Results:

6a) 72% of participants were migrants: what was the definition of a migrant and what was their origin?

6b) Which cut-off was used to define a positive PCR test result and how was this cut-off determined?

6c) 25% of participants had contact with a Mpox case, how was this contact defined?

6d) If a questionnaire was used, then this questionnaire should be made available as a supplement.

6e) The authors mention that their results show the benefits of working together with community organizations. Please explain in the text how the authors collaborated with community organizations.

7) Some sentences/paragraphs do not follow a logic flow. Please rephrase:

7a) "Although, sexual transmission by means of semen has not been ruled out, some authors suggest that rather than the respiratory route, local inoculation by close skin-to-skin contact during sexual activity is the dominant transmissibility mode of MPXV, in non-endemic Mpox countries".

7b) "Eight anal samples yielded an inconclusive PCR result, as no human DNA (myostatine gene) was detected"

7c) "It is of note that viable MPXV viruses were obtained from individuals reporting symptoms, although one had no symptoms before testing"

7d) "Tarin et al⁶ proposed skin-to-skin contact rather than the respiratory route as the dominant mode of MPXV transmission outside countries where the virus is endemic based on the history of sexual exposure, predominant anogenital skin lesions, and higher viral loads in skin than throat swabs. Here, we confirm that pharyngeal swabs allow for the isolation of viable viruses. As, MPXV has been isolated from semen^{18–20} these findings further corroborates the role of sexual transmission of MPXV during the 2022 outbreak."

Table 1, 2 and 4:

8) As mentioned, I recommend to remove all p-values

9) I recommend to reduce the length of the tables, for example by combining some categories.

Table 4:

10) Grouping of results by test-positive and test-negative patients is not meaningful for these outcomes

11) Explain "if necessary" in "Preferred location for repeat testing if necessary"?

NCOMMS-23-05438-T

ASYMPTOMATIC MONKEY POX VIRUS INFECTION: A SELF-SAMPLING SCREENING INTERVENTION
ADRESSED TO GAY, BISEXUAL AND OTHER MEN WHO HAVE SEX WITH MEN AND TRANS WOMEN
IN SPAIN

Reviewer's Comments:

Reviewer #1 (Remarks to the Author):

Comment: Line 116: a "dot" is missing after « migrants »

Answer: Thank you very much for this point. It has been corrected in the new version of the manuscript.

Comment: I have a major concern:

Line 104: n the present study, we aimed (i) to assess the prevalence of MPXV infection among asymptomatic highly exposed GBMSM and trans women (TW) who were recruited in a community-based centre in Barcelona

Line 172: Three (3/6) participants reported the following symptoms before testing: a swollen inguinal lymph node, fever, exhaustion and a skin lesion but none of them connected these symptoms with MPXV infection

These 3 individuals are NOT asymptomatic. Such symptoms can be prodrom symptoms of mpox infection, therefore these individuals should not be counted as "asymptomatic". If authors want to keep them in the analysis, then they should modify their objective and rephrase it: to assess the prevalence of MPXV infection among asymptomatic individuals and in individuals with mild unrecognized Mpox symptoms.

Answer: Thank you for your comments. After careful consideration and discussion, following your recommendation, we have decided to modify the objective of the study. Although the original aim was to assess the prevalence of MPXV infection among asymptomatic highly exposed GBMSM and trans women, some participants reported symptoms that were compatible with MPXV infection, but they did not connect these symptoms to the infection. We chose not to exclude the participants who presented mild unrecognized symptoms from the study due to their relevance in the transmission of the infection. In the revised version of the manuscript, we have provided a justification for the decision of not excluding individuals with unrecognized symptoms and estimated the overall prevalence of MPXV infection, as well as the prevalence of MPXV infection taking into account only those participants without symptoms. We hope that this clarifies our approach and addresses your concern.

The following justification has been included in the Discussion section of the new version of the manuscript:

"The decision was made not to exclude participants who exhibited mild unrecognized symptoms from the study, given the importance of individuals with undetected symptoms in the transmission of the infection".

Comment: Line 201: Three out of the 7 participants who tested positive for MPXV through our study did not have any symptom before testing, while three of them had symptoms that were easily confused with STIs that cause skin rashes or mucosal lesions, such as genital herpes, syphilis, acuminated condyloma and chancroid among others

This is very confusing. Authors do state that mpox is an STI, or is more likely transmitted via the sexual route, but when symptoms suggestive of STI are reported, authors state that such symptoms are not recognized as possibly related to mpox....

Answer: We thank the reviewer for their comment and pointing out a confusion caused by the wording in the manuscript. We agree that MPXV can be transmitted sexually and can present with symptoms that are suggestive of other STIs, including genital herpes, syphilis, acuminated condyloma, and chancroid. However, in our study, we focused on assessing the prevalence of MPXV infection among asymptomatic highly exposed GBMSM and trans women who did not report any symptoms that were specifically associated with MPXV infection. While some participants did report symptoms that were suggestive of STIs, including those mentioned, they did not connect these symptoms with MPXV infection. We acknowledge that these symptoms could potentially be related to MPXV infection, and we have included this information in the revised version of the manuscript. We hope that this clarifies any confusion and provides a more accurate representation of our findings.

The following texts are included in the revised version of the manuscript:

In the results section: “Three (3/6) participants reported the following symptoms before testing: a swollen inguinal lymph node, fever, exhaustion and a skin lesion and none of those participants connected these symptoms with MPXV infection”.

In the Discussion section: “Three out of the 7 participants who tested positive for MPXV through our study did not have any symptom before testing, while three of them had symptoms that were easily confused with STIs that cause skin rashes or mucosal lesions, such as genital herpes, syphilis, acuminated condyloma and chancroid among others; or even COVID-19 (which causes fever and exhaustion). However, even MPXV is more likely transmitted via the sexual route and these symptoms could potentially be related to MPXV infection, those participants did not relate these symptoms with Mpox”.

Comment: Line 206: We were able to identify replication-competent virus particles in three out of six (3/6) MPXV positive individuals. One of them did not present symptoms before testing and the other two did not recognize their symptoms as indicative of MPXV infection.

◇ again, 2 individuals had STI symptoms. If they did not recognize such symptoms as possibly related to mpox infection, they did seek for medical examination as they did come to the sexual health clinic. So this cannot be considered as a study of prevalence of mpox among “asymptomatic” individuals.

Answer: Thank you for your comment. We appreciate your concern regarding the categorization of participants as asymptomatic. We acknowledge that some participants reported symptoms that were suggestive of STIs, including two individuals who tested positive for MPXV and did not recognize their symptoms as indicative of MPXV infection. However, it is important to note that

these participants were not recruited while seeking medical examination and we have no information if they did go to a sexual health clinic. The participants were recruited at the collaborating community center that, apart from offering the rapid HIV test, provides accompaniment services for LGBTBI+ people, sex workers, people with HIV, among others. We agree with you that although these participants did not connect their symptoms to MPXV infection, they can not be considered asymptomatic and it has been reflected in the manuscript. The decision was made not to exclude those participants who exhibited mild unrecognized symptoms from the study, given the importance of individuals with undetected symptoms in the transmission of the infection. Therefore, we have revised the description of our study to better reflect this aspect, and we have provided a justification for our approach in the revised manuscript. We hope that this clarifies our methodology and addresses your concern.

Comment: Another concern is on the way the results were given to participants. Were they told to reduce their sexual activity and to notify their partners? The sentence line 210 and 335 are not clear on this matter.

Answer: We understand your concern about this issue. All participants with a positive result were told to isolate themselves and sexual abstinence and partner notification were recommended, following the current protocol in Catalonia (Spain). This explanation is included in the Methods section.

Comment: Were they followed-up? Was the test redone for confirmation? Line 335 it is mentioned that they were asked to go to their GP or to a sexual health clinic “for confirmation”. How many of them had a confirmatory sample? An what was the result of the confirmatory sample?

Answer: The authors acknowledge the reviewer's concern regarding the confirmation process for positive results. There was not a confirmation process of the positive results. None of the participants with a positive result confirmed their result in a clinical setting, which was an error in the manuscript. Instead, participants were instructed to follow up with their GPs or STI clinics for further evaluation.

It has been corrected in the text: “All participants with a positive result for MPXV infection were instructed to attend their General Practitioner (GP) or a STI Clinic for follow-up and were advised on isolation measures and sexual abstinence”.

Comment: The discussion section is too long and mixes scientific virological findings with socio-behavioral assumptions and on the acceptability/feasibility on self-sampling, that are not supported by the results presented herein and/or were not part of the objectives of the present study.

Answer: Thank you very much for your feedback on our manuscript. While we appreciate your input, we respectfully disagree that the discussion section is too long. Our aim was to provide a comprehensive analysis of the findings and their implications, and we believe that the length of the discussion is appropriate for the scope and complexity of the study. That being said, we carefully reviewed the Discussion section to ensure that every point is essential and well-supported. On the other hand, we consider that the integration of scientific virological findings

with socio-behavioral assumptions, acceptability, and feasibility of new testing strategies is crucial in public health research because it allows for the development of effective and practical testing strategies that are acceptable to the public. Understanding the virology of a virus and the modes of transmission is important, but it is also essential to consider how people will react to new testing strategies and whether they will be willing to participate. Furthermore, it is important to take into account the cultural and social factors that may influence the acceptability and feasibility of new testing strategies. It has been seen that MPXV testing may be stigmatized or perceived as intrusive, which could impact participation rates and the effectiveness of the testing program. By integrating both scientific and socio-behavioral perspectives, public health researchers can develop testing strategies that are not only effective but also acceptable and feasible for the populations they are intended to serve. This, in turn, can help to improve the uptake and impact of testing programs and ultimately contribute to the control and prevention of Mpox.

Following the recommendations of the reviewers, the following sentences that imply that this study affected behavior of participants and/or transmission from participants as this was not formally tested in the study have been removed from the Discussion section:

“As positive MPXV cases were not aware of their infection, without our self-sampling intervention they would not have attended a health care setting, or get diagnosed, and consequently they would not have self-isolated and we would not have carried out contact tracing. Therefore, if they had not known their infection status and as they presented no symptoms or did not recognize them, these individuals would have continued to spread the infection unknowingly.”

Comment: Conclusion line 270: We also demonstrate that MPXV symptoms can overlap and be confused with other diseases, such as other STIs. ◊ this has been previously reported and published by other groups such as the US CDC (June – July 2022).

Answer: Thank you very much for this point. The suggested reference has been included in the Discussion section:

“Three out of the 7 participants who tested positive for MPXV through our study did not have any symptom before testing, while three of them had symptoms that were easily confused with STIs that cause skin rashes or mucosal lesions, such as genital herpes, syphilis, acuminate condyloma and chancroid among others, as previously described¹⁸”.

18: Philpott D, Hughes CM, Alroy KA, Kerins JL, Pavlick J, Asbel L, Crawley A, Newman AP, Spencer H, Feldpausch A, Cogswell K, Davis KR, Chen J, Henderson T, Murphy K, Barnes M, Hopkins B, Fill MA, Mangla AT, Perella D, Barnes A, Hughes S, Griffith J, Berns AL, Milroy L, Blake H, Sievers MM, Marzan-Rodríguez M, Tori M, Black SR, Kopping E, Ruberto I, Maxted A, Sharma A, Tarter K, Jones SA, White B, Chatelain R, Russo M, Gillani S, Bornstein E, White SL, Johnson SA, Ortega E, Saathoff-Huber L, Syed A, Wills A, Anderson BJ, Oster AM, Christie A, McQuiston J, McCollum AM, Rao AK, Negrón ME; CDC Multinational Monkeypox Response Team. Epidemiologic and Clinical Characteristics of Monkeypox Cases - United States, May 17-July 22, 2022. *MMWR Morb Mortal Wkly Rep.* 2022 Aug 12;71(32):1018-1022. doi: 10.15585/mmwr.mm7132e3. PMID: 35951487; PMCID: PMC9400536.

In addition, we agree with you that this was previously described, and the conclusion has been modified as follow (changing “confirming” by “demonstrating”): “We also confirmed that MPXV symptoms can overlap and be confused with other diseases, such as other STIs”.

Reviewer #2 (Remarks to the Author)

This is a very interesting paper that focuses on asymptomatic infections of Mpox. It would be interesting to understand possible further transmission although I understand this may not be within the scope of the paper. I have some questions and minor amendments to the manuscript below:

Methods

Comment: More details on how the authors arrive at a sample size of 113 participants? How many were excluded and the dropout rate. What biases did this introduce.

Answer: Thank you very much for your feedback. We managed to recruit a sample of 113 participants through intermittent dissemination campaigns on Instagram, Facebook, WhatsApp and the community centre website. The campaigns indicated the possibility of get tested for MPXV through a self-sampling intervention free of charge at the headquarters of the collaborating community centre. Participants with eligible criteria were invited to attend the collaborating community centre to get tested for MPXV. No participant was excluded in the study. More detailed information on the content of the dissemination campaigns has been included in the Methods section. Regarding the potential introduced biases, see the answer to the third comment.

Comment: Was sample size a consideration in study design?

Answer: Yes, It was. It was expected to include 150 participants in the study. A random sample of 142 individuals was established as sufficient to estimate, with 95% confidence and an accuracy of +/- 2 percentage units, a population percentage that was expected to be around 1.5% (De Baetselier I, Van Dijck C, Kenyon C, Coppens J, Michiels J, de Block T, Smet H, Coppens S, Vanroye F, Bugert JJ, Giral P, Zange S, Liesenborghs L, Brosius I, van Griensven J, Selhorst P, Florence E, Van den Bossche D, Ariën KK, Rezende AM, Vercauteren K, Van Esbroeck M; ITM Monkeypox study group. Retrospective detection of asymptomatic monkeypox virus infections among male sexual health clinic attendees in Belgium. *Nat Med.* 2022 Nov;28(11):2288-2292. doi: 10.1038/s41591-022-02004-w. Epub 2022 Aug 12. PMID: 35961373; PMCID: PMC9671802.). Finally, 113 individuals participated in the study.

We have also included an explanation of why only 113 participants were recruited in the new version of the manuscript:

Methods section:

“After the start of vaccination and the decrease in new cases of mpox in Spain, the interest of the target population in a screening intervention dropped considerably and it was impossible to reach the expected n (150).”

We have also discussed how this affects study aim/outcome:

Study limitations (Discussion section): Although the precision of the estimates decreased due to a reduction in the expected number of participants, the confidence intervals have remained significant but wider because the obtained prevalence was higher than the prevalence used to calculate the sample size.

Comment: What biases were present during recruitment?

Answer: The authors acknowledge the reviewer's concern regarding the potential biases present during the recruitment. The collaborating center work mainly with gay, bisexual and other men who have sex with men and trans women. They have a specific program for sex workers and most of them are migrant. This could explain why among the studied population there was a great percentage of migrants and sex workers. This could be a bias and the first limitation reported by the authors in the article is: "The study population is not representative of GBMSM and TW in Catalonia as we used an opportunistic sample". The following text has been added: "However, it is worth noting that the sample had a notably high proportion of migrants and sex workers, likely due to the involvement of a collaborating center with a specific program for this population. While this may introduce some bias, we did not observe significant differences in the number of sexual partners in the last 30 days or the history of exchanging sex for money, gifts, or favors between participants with positive and negative MPXV infection results".

Comment: "participants had received the Mpox vaccine in their childhood or in the previous 12 months, respectively" - do the authors mean the smallpox vaccine in childhood here?

Answer: Yes, the authors referred to the smallpox vaccine. It has been clarified in the new version of the manuscript: "History of small pox and Mpox vaccination".

Comment: "38 (33.63%) had had sex in exchange for money, gifts or favours" - this suggests a recruitment bias that would be relevant to discuss.

Answer: The authors acknowledge the potential for recruitment bias related to the proportion of participants who reported exchanging sex for money, gifts, or favors. However, it is important to note that this high percentage may reflect the population served by the collaborating center that recruited the study participants, which had a specific program for sex workers and migrants. We have added a discussion of this potential bias in the revised manuscript (See Limitations of the study) to provide greater transparency and contextualization for the findings.

Comment: Some high CT values "36.79" that would perhaps warrant a retest per CDC advice "If you obtain a high Ct value (generally ~34 or higher), CDC recommends to immediately re-extract and re-test to ensure there was no cross-contamination."(https://www.cdc.gov/locs/2022/08-23-2022-Lab-Advisory-Monkeypox_Virus_Testing_Considerations_Prevent_False_Positive_Test_Results.html).

Worth including this in the limitations.

Answer: We have reviewed all the raw data and PCR curves in all samples including those with high Ct values that are also recommended by the CDC. In samples with high Ct re-extract and re-test was not performed due to the small volume of original sample to carry out all the experiments.

We have included the following text in the Study limitations (Discussion Section):

“Fourthly, due to the high sensitivity of PCR techniques, there is the possibility of obtaining false positive results. In this situation, Ct values are high ($Ct > 34$) and are associated with low viral loads and the laboratory should re-extract and re-amplify the original sample or a new sample requested if this is not possible²⁹. In our study, there were three samples with high Cts (64 pharynx; 72 anal; 81 pharynx). Re-testing these samples was not possible due to the low volume left. However, the PCR performed after the viral culture for MPXV was positive for one sample (sample 72). Previous observations have shown that pharyngeal and anal samples exhibit higher Cts in comparison to samples obtained from lesions^{4,7}. Furthermore, good practices were followed to prevent contamination and erroneous diagnoses, and negative controls produced the expected result. Moreover, the study's conclusions rely on samples in which the ability of MPXV to replicate has been verified and validated using diverse and independent methods.”

Comment: Line 422 "MPXV infection prevalence was estimated by calculating the proportion of individuals with a positive result over the total of individuals with a returned and valid sample. Confidence interval of 95% was calculated." - This needs expanding. Are they binomial confidence intervals? If so what method?

Answer: Thank you for your comment and for bringing attention to the need for further clarification regarding the calculation of confidence intervals for MPXV infection prevalence. We used the binomial method to estimate 95% confidence intervals for the proportion of individuals with positive test results. We have included this information in the revised manuscript to provide more transparency and to aid the reproducibility of our study.

Comment: A suitable alternative method that authors may want to consider for investigating the characteristics of the patients with mpox could be logistic regression.

Answer: Thank you for your comment and suggestion regarding the use of logistic regression for investigating the characteristics of patients with mpox. We appreciate your input and agree that logistic regression could be a useful alternative method for this analysis. However, given the relatively small number of positive cases (only seven individuals), we ultimately concluded that the sample size was insufficient to fit a model, particularly for covariates with more than two categories or those with unbalanced proportions. Furthermore, following the recommendation of other reviewer we removed all the statistical analyses that were clearly not driven by a prior hypothesis.

Reviewer #3 (Remarks to the Author)

The authors performed a study among the clients of a community centre where STI testing was offered. Eligible participants were individuals at high risk for mpox according to a set of criteria. Participants were given a self-sampling kit for anorectal and pharyngeal sampling and surveys about the acceptability of self-sampling were performed concomitantly. The authors found 7 mpox infections among 113 individuals tested, and 6 of them did not report symptoms at the time of testing, nor did 2 of them in the subsequent 21 days. The authors ascertained the presence of viable virus by culture in 4 cases of which 1 did not report any symptoms at the time of testing. The results of this study are in line with the findings of previous studies that found undiagnosed mpox infections, of which the authors cite two.

I'd like to compliment the study team for their achievement to recruit >100 participants among a population that may be difficult to engage in clinical studies, and for their efforts to confirm the results of MPCV culture by several techniques.

Major comments:

Comment: 1)The main aim of the study was “to assess the prevalence of MPXV infection among asymptomatic highly exposed GBMSM and trans women”. Please provide more information in the main text and in the abstract about the targeted population and the way of sampling and data collection so that the reader can obtain an accurate of the population to which the prevalence can be generalised.

Answer: Thank you for your comments and for bringing attention to the need for further clarification about the targeted population and the way of sampling and data collection. In response to the reviewer's comments, we have made several additions to the Methods section of the manuscript to provide more information on recruitment and data collection. Specifically, we have clarified that information on participant behavior was collected via a self-administered paper survey, and that data related to symptoms among those who tested positive for MPXV infection was obtained through a phone survey conducted by the field coordinator 21 days after receiving the test result. Furthermore, we have expanded the description of the targeted population to include additional socio-demographic information, particularly for immigrant participants. Two paragraphs now summarize the information collected, including details on behavior and possible exposure to MPXV. Finally, we have also extended the information on the targeted population in the abstract to provide a more comprehensive overview of our study

Comment: What was the study design and was this a prospective or retrospective study?

Answer: We performed a transversal prospective non-randomized study. The word “prospective” has been added in the revised version of the manuscript. See Methods section.

Comment: When was the study performed with respect to the mpox epidemic?

Answer: The study was performed between August and October 2022 (See first paragraph of Results section). The epidemiological curve of Mpox outbreak in Spain showed that the first case was reported in April 26th 2022, the peak of the outbreak was in July 2022 and the cases considerably decreased in September (coinciding with vaccination campaign in the country)

(<https://www.isciii.es/QueHacemos/Servicios/VigilanciaSaludPublicaRENAVE/EnfermedadesTransmisibles/Documents/archivos%20A-Z/MPOX/SITUACION%20EPIDEMIOLOGICA%20DE%20LOS%20CASOS%20DE%20VIRUELA%20DE%20MONO-07022023.pdf>); so the study was performed shortly after the peak of the epidemics. This information has been added in the Methods Section.

Comment: Who was eligible and who was not?

Answer: The study targeted two different key populations: GBMSM and TW. The inclusion criteria were: Being over 18 years old, with no symptoms of MPXV infection and considered at high risk of contracting Mpox. High risk was defined as: GBMSM and TW who are sex workers and/or chemsex users and/or who practice group sex and/or are HIV positive or are PrEP users. See Methods Section.

Comment: How and where were participants recruited?

Answer: We have expanded the information included in the method section about how the recruitment was performed: The study was disseminated through Instagram, Facebook, WhatsApp and the community centre website via intermittent campaigns. The campaigns indicated the possibility of get tested for MPXV through a self-sampling intervention free of charge at the headquarters of the collaborating community centre. Participants with eligible criteria were invited to attend the collaborating community centre to get tested for MPXV. No participant was excluded in the study. The community centre staff briefly explained the project to potential participants and obtained the signed informed consent on paper.

Comment: How and when were their symptoms and signs evaluated?

Answer: Three weeks after testing, all participants with a positive result were contacted by phone the field coordinator to enquire if they had had any symptom before testing them and within the following 21 days. No clinical assessments were conducted at the time of recruitment or after testing. Symptoms therefore, were self-reported. This has been clarified in the Methods Section.

We also included the following sentence in the Methods Section: “The follow-up period was set at 21 days because the incubation period for Mpox is between 5 and 21 days.”

Comment: How were clinical and other (survey?) data collected?

Answer: Participants answered a self-completed paper survey on sociodemographic characteristics and behaviour before self-collecting their samples. The survey has been included as a Supplementary Material in the new version of the manuscript. Data related to symptoms on those participants with a positive result was collected through a survey performed through a phone call by the field coordinator after 21 days of receiving test result.

We used RedCAP (REDCap systems, Vanderbilt University, US) to collect data of participants and create an *ad hoc* online data base. We carried out the data entry of the survey data at the coordinating centre.

Comment: Two participants reported a swollen lymph node and a skin lesion, so according to the current inclusion criteria, these patients may not have been eligible?

Answer: Thank you for your comment. Although the original aim of the study was to assess the prevalence of MPXV infection among asymptomatic highly exposed GBMSM and trans women, some participants reported symptoms that were compatible with MPXV infection, but they did not connect these symptoms to the infection. We chose not to exclude the participants who presented mild unrecognized symptoms from the study due to their relevance in the transmission of the infection. In the revised version of the manuscript, we have provided a justification for this decision and estimated the overall prevalence of MPXV infection, as well as the prevalence of MPXV infection among those participants without symptoms. We hope that this clarifies our approach and addresses your concern.

Comment: Even though it is not clear from the text, I suppose that this information was collected retrospectively, in hindsight? If no thorough clinical assessment was done at the time of recruitment, I would suggest to rephrase the aim of the study and the primary outcome.

Answer: We appreciate the reviewer's feedback. To clarify, data related to symptoms on those participants with a positive result for Mpox was collected retrospectively through a survey performed through a phone call by the field coordinator after 21 days of receiving test result. No clinical assessments were conducted at the time of recruitment and after testing. This information has been included in the Methods section.

Thank you very much for your suggestion of rephrasing the aim of the study and the primary outcome. After careful consideration and discussion, and following your recommendation, we have decided to modify the objective of the study. As we commented above, in spite of the initial objective being the evaluation of the prevalence of MPXV infection among asymptomatic highly exposed GBMSM and trans women, certain participants reported symptoms compatible with MPXV infection, but were not aware of the connection between these symptoms and the infection. We decided not to exclude these participants with mild unrecognized symptoms from the study, considering their importance in the transmission of the infection. In the updated manuscript, we have modified the original aim of the study and we have explained the reasoning behind the decision of not excluding individuals with unrecognized symptoms and calculated both the overall prevalence of MPXV infection and the prevalence of MPXV infection among participants without symptoms.

Comment: 2) The statistical analyses in this study are clearly not driven by a prior hypothesis and many of the comparisons are not meaningful, such as the comparison of acceptability of self-testing between test-positive and test-negative cases. Therefore, I recommend to drop all statistical analysis from the manuscript, or only perform those tests that are hypothesis-driven and corrected for multiple testing.

Answer: The authors agree with the reviewer and following their recommendation all statistical analyses have been removed from the manuscript.

Comment: 3) Which efforts have been done to exclude that positive results in cases without clear mpox symptoms were caused by contamination? Was infection confirmed on a second sample or through serology?

Answer: All samples were handled in a type II biological safety cabinet. For the extraction, the work was done in automatic processors that have a protocol to avoid the formation of aerosols. In addition, preventive cleaning maintenance was carried out on said automatic processors and surfaces of the work areas. In each RUN of samples a negative control was added and in all of them the result was DNA monkey not detected. In short, good molecular biology laboratory practices have been followed.

The results obtained by PCR with a second extraction were not confirmed, but in all cases the virological culture was performed from an unmanipulated aliquot of the original sample.

Serology was not performed as it was not foreseen in the study.

Minor comments:

Comment: 4) Introduction, study aim II) “to assess the potential transmissibility of MPXV”. I recommend to change wording to avoid the word “transmissibility” because this study was not designed to evaluate MPXV transmission.

Answer: Following the recommendation of the reviewer the authors changed the second objective of the study: To assess the presence of replication-competent particles of MPXV.

Comment: 5) Abstract: I suggest to not mention viral loads here

Answer: The authors followed the recommendation of the reviewer and have removed “viral loads” from the abstract.

6) Results:

Comment: 6a) 72% of participants were migrants: what was the definition of a migrant and what was their origin?

Answer: Migrant was defined as being born in a country different from Spain. It has been included in the text their origin.

Comment: 6b) Which cut-off was used to define a positive PCR test result and how was this cut-off determined?

Answer: The assay detection limit was used. Values greater than or equal of detection limit were considered positive.

Comment: 6c) 25% of participants had contact with a Mpox case, how was this contact defined?

Answer: We asked in the survey if they have had contact with a case of Mpox in the last 30 days. We also asked about kind of contact: At work (puncture, laboratory work, contact with potentially contaminated material, healthcare professional without PPE), person I take care of. Sexual contact (groping, masturbation, oral sex, vaginal or anal penetration with or without ejaculation). Shared food, utensils, or dishes. Shared clothes. Towels or bed linen shared at home or elsewhere. Having gone on a trip together. Shared bathrooms (sinks, showers) either at home or elsewhere. Physical contact (face to face, kissing, shaking hands, hugging...). Other.

We have included this information in the revised version of the manuscript, see Method section.

Comment: 6d) If a questionnaire was used, then this questionnaire should be made available as a supplement.

Answer: We have made available the questionnaire as a supplementary material.

Comment: 6e) The authors mention that their results show the benefits of working together with community organizations. Please explain in the text how the authors collaborated with community organizations.

Answer: The collaborating community organization participated in the conceptualization of the study, elaboration of the messages, dissemination of the intervention and facilitating access to the target population. It has been added in the new version of the manuscript. See Methods section (Ethical Considerations).

Comment: 7) Some sentences/paragraphs do not follow a logic flow. Please rephrase:

7a) "Although, sexual transmission by means of semen has not been ruled out, some authors suggest that rather than the respiratory route, local inoculation by close skin-to-skin contact during sexual activity is the dominant transmissibility mode of MPXV, in non-endemic Mpox countries".

Answer: Following the suggestion of the reviewer, the following sentences have been rephrased: "Some authors suggest that, instead of respiratory transmission, the dominant transmissibility mode of MPXV, in non-endemic Mpox countries, is local inoculation by close skin-to-skin contact during sexual activity; although, sexual transmission by means of semen has not been ruled out".

Comment: Please rephrase: 7b) "Eight anal samples yielded an inconclusive PCR result, as no human DNA (myostatin gene) was detected"

Answer: The text has been modified: "The PCR test produced an inconclusive result for 8 anal samples due to the absence of human DNA (myostatin gene). This lack of detection could have been caused by the presence of inhibitors or a potential error during the sample collection process".

Comment: Please rephrase: 7c) "It is of note that viable MPXV viruses were obtained from individuals reporting symptoms, although one had no symptoms before testing"

Answer: Following the recommendations of the reviewer, these sentences have been rephrased: "It's worth noting that viable MPXV viruses were obtained from individuals who reported symptoms. Two of these individuals reported experiencing mild symptoms prior to testing, although they didn't relate them to MPX, while one individual didn't display any symptoms prior to testing".

Comment: Please rephrase: 7d) "Tarin et al6 proposed skin-to-skin contact rather than the respiratory route as the dominant mode of MPXV transmission outside countries where the virus is endemic based on the history of sexual exposure, predominant anogenital skin lesions,

and higher viral loads in skin than throat swabs. Here, we confirm that pharyngeal swabs allow for the isolation of viable viruses. As, MPXV has been isolated from semen^{18–20} these findings further corroborates the role of sexual transmission of MPXV during the 2022 outbreak.”

Answer: Tarin et al. (6) proposed that skin-to-skin contact, rather than the respiratory route, is the dominant mode of MPXV transmission outside endemic countries. This was based on the history of sexual exposure, predominant anogenital skin lesions, and higher viral loads in skin than throat swabs. As, MPXV has been isolated from semen (18–20) these findings further corroborates the role of sexual transmission of MPXV during the 2022 outbreak. Our findings show that viable viruses can be isolated from pharyngeal swabs.

Table 1, 2 and 4:

Comment: 8) As mentioned, I recommend to remove all p-values

Answer: Following the reviewer’s recommendation, all p values have been removed.

Comment: 9) I recommend to reduce the length of the tables, for example by combining some categories.

Answer: Thank you for your suggestion. However, we believe that presenting the data in a detailed and transparent way is important for readers to understand the study results. We have considered combining some categories to make the tables more concise, but we found that it would have limited the information provided about the studied population. After removing the p-values the tables are easier to read.

Table 4:

Comment: 10) Grouping of results by test-positive and test-negative patients is not meaningful for these outcomes.

Answer: The authors appreciate the suggestion of the reviewer. The table has been simplified.

Comment: 11) Explain “if necessary” in “Preferred location for repeat testing if necessary”?

Answer: Thanks for bringing attention to this issue. If necessary has been substituted by “if it was necessary to repeat the test”

Reviewers' Comments:

Reviewer #1:

Remarks to the Author:

Authors have addressed my comments and have modified the objective of the study by clearly mentioning that they included asymptomatic and participants with mild/unrecognized symptoms. However, my comment regarding the discussion section - i.e. the discussion section is too long and mixes scientific virological findings with socio-behavioral assumptions and on the acceptability/feasibility on self-sampling, that are not supported by the results presented herein and/or were not part of the objectives of the present study - has not been addressed as authors state that they disagree with my comment.

Reviewer #2:

Remarks to the Author:

Reviewer #2 (Remarks to the Author)

This is a very interesting paper that focuses on asymptomatic infections of Mpox. It would be interesting to understand possible further transmission although I understand this may not be within the scope of the paper. I have some questions and minor amendments to the manuscript below:

Methods

Comment: More details on how the authors arrive at a sample size of 113 participants? How many were excluded and the dropout rate. What biases did this introduce.

Answer: Thank you very much for your feedback. We managed to recruit a sample of 113 participants through intermittent dissemination campaigns on Instagram, Facebook, WhatsApp and the community centre website. The campaigns indicated the possibility of getting tested for MPXV through a self-sampling intervention free of charge at the headquarters of the collaborating community centre. Participants with eligible criteria were invited to attend the collaborating community centre to get tested for MPXV. No participant was excluded in the study. More detailed information on the content of the dissemination campaigns has been included in the Methods section. Regarding the potential introduced biases, see the answer to the third comment.

Comment 2: Thank you for including this context in the paper.

Comment: Was sample size a consideration in study design?

Answer: Yes, it was. It was expected to include 150 participants in the study. A random sample of 142 individuals was established as sufficient to estimate, with 95% confidence and an accuracy of +/- 2 percentage units, a population percentage that was expected to be around 1.5% (De Baetselier I, Van Dijck C, Kenyon C, Coppens J, Michiels J, de Block T, Smet H, Coppens S, Vanroye F, Bugert JJ, Giral P, Zange S, Liesenborghs L, Brosius I, van Griensven J, Selhorst P, Florence E, Van den Bossche D, Ariën KK, Rezende AM, Vercauteren K, Van Esbroeck M; ITM Monkeypox study group. Retrospective detection of asymptomatic monkeypox virus infections among male sexual health clinic attendees in Belgium. *Nat Med.* 2022 Nov;28(11):2288-2292. doi: 10.1038/s41591-022-02004-w. Epub 2022 Aug 12. PMID: 35961373; PMCID: PMC9671802.). Finally, 113 individuals participated in the study.

Comment 2: How did you establish this random sample was sufficient? I would include further context.

We have also included an explanation of why only 113 participants were recruited in the new version of the manuscript:

Methods section:

"After the start of vaccination and the decrease in new cases of mpox in Spain, the interest of the target population in a screening intervention dropped considerably and it was impossible to reach the expected n (150)."

Comment 2: I would not use the word "impossible" here but rather difficult.

We have also discussed how this affects study aim/outcome:

Study limitations (Discussion section): Although the precision of the estimates decreased due to a reduction in the expected number of participants, the confidence intervals have remained significant but wider because the obtained prevalence was higher than the prevalence used to calculate the sample size.

Comment: What biases were present during recruitment?

Answer: The authors acknowledge the reviewer's concern regarding the potential biases present during the recruitment. The collaborating center work mainly with gay, bisexual and other men who have sex with men and trans women. They have a specific program for sex workers and most of them are migrant. This could explain why among the studied population there was a great percentage of migrants and sex workers. This could be a bias and the first limitation reported by the authors in the article is: "The study population is not representative of GBMSM and TW in Catalonia as we used an opportunistic sample". The following text has been added: "However, it is worth noting that the sample had a notably high proportion of migrants and sex workers, likely due to the involvement of a collaborating center with a specific program for this population. While this may introduce some bias, we did not observe significant differences in the number of sexual partners in the last 30 days or the history of exchanging sex for money, gifts, or favors between participants with positive and negative MPXV infection results".

Comment 2: Is the sample size adequate to say "we did not observe significant differences in the number of sexual partners in the last 30 days or the history of exchanging sex for money, gifts, or favors between participants with positive and negative MPXV infection results" with "significant" here implied as a statistical term.

Comment: "participants had received the Mpox vaccine in their childhood or in the previous 12 months, respectively" - do the authors mean the smallpox vaccine in childhood here?

Answer: Yes, the authors referred to the smallpox vaccine. It has been clarified in the new version of the manuscript: "History of small pox and Mpox vaccination".

Comment 2: Did any participants receive the mpox vaccine or was it all smallpox?

Comment: "38 (33.63%) had had sex in exchange for money, gifts or favours" - this suggests a recruitment bias that would be relevant to discuss.

Answer: The authors acknowledge the potential for recruitment bias related to the proportion of participants who reported exchanging sex for money, gifts, or favors. However, it is important to note that this high percentage may reflect the population served by the collaborating center that recruited the study participants, which had a specific program for sex workers and migrants. We have added a discussion of this potential bias in the revised manuscript (See Limitations of the study) to provide greater transparency and contextualization for the findings.

Comment 2: Thank you for including this context

Comment: Some high CT values "36.79" that would perhaps warrant a retest per CDC advice "If you obtain a high Ct value (generally ~34 or higher), CDC recommends to immediately re-extract and re-

test to ensure there was no cross-contamination."(https://www.cdc.gov/locs/2022/08-23-2022-Lab-Advisory-Monkeypox_Virus_Testing_Considerations_Prevent_False_Positive_Test_Results.html). Worth including this in the limitations.

Answer: We have reviewed all the raw data and PCR curves in all samples including those with high Ct values that are also recommended by the CDC. In samples with high Ct re-extract and re-test was not performed due to the small volume of original sample to carry out all the experiments.

We have included the following text in the Study limitations (Discussion Section):

"Fourthly, due to the high sensitivity of PCR techniques, there is the possibility of obtaining false positive results. In this situation, Ct values are high ($Ct > 34$) and are associated with low viral loads and the laboratory should re-extract and re-amplify the original sample or a new sample requested if this is not possible²⁹. In our study, there were three samples with high Cts (64 pharynx; 72 anal; 81 pharynx). Re-testing these samples was not possible due to the low volume left. However, the PCR performed after the viral culture for MPXV was positive for one sample (sample 72). Previous observations have shown that pharyngeal and anal samples exhibit higher Cts in comparison to samples obtained from lesions^{4,7}. Furthermore, good practices were followed to prevent contamination and erroneous diagnoses, and negative controls produced the expected result. Moreover, the study's conclusions rely on samples in which the ability of MPVX to replicate has been verified and validated using diverse and independent methods."

Comment 2: Thank you for including this context

Comment: Line 422 "MPXV infection prevalence was estimated by calculating the proportion of individuals with a positive result over the total of individuals with a returned and valid sample. Confidence interval of 95% was calculated." - This needs expanding. Are they binomial confidence intervals? If so what method?

Answer: Thank you for your comment and for bringing attention to the need for further clarification regarding the calculation of confidence intervals for MPXV infection prevalence. We used the binomial method to estimate 95% confidence intervals for the proportion of individuals with positive test results. We have included this information in the revised manuscript to provide more transparency and to aid the reproducibility of our study.

Comment 2: Thank you for including this. I would still include the method used to calculate the binomial uncertainty.

Comment: A suitable alternative method that authors may want to consider for investigating the characteristics of the patients with mpox could be logistic regression.

Answer: Thank you for your comment and suggestion regarding the use of logistic regression for investigating the characteristics of patients with mpox. We appreciate your input and agree that logistic regression could be a useful alternative method for this analysis. However, given the relatively small number of positive cases (only seven individuals), we ultimately concluded that the sample size was insufficient to fit a model, particularly for covariates with more than two categories or those with unbalanced proportions. Furthermore, following the recommendation of other reviewer we removed all the statistical analyses that were clearly not driven by a prior hypothesis.

Comment 2:I think this is a fair justification.

Reviewer #3:

Remarks to the Author:

The manuscript title is not grammatically correct.

Likewise for some of the revisions in the text.

Abstract:

- I suggest to simplify the first sentence as "we aimed to assess the prevalence of monkeypox virus (MPXV) among gay, bisexual, and other men who have sex with men and trans women (TW) without symptoms or with unrecognized Mpox symptoms, using a self-sampling strategy"
- 113 individuals participated: 89 men + 17 TW = 106 => what was the gender of the remaining 7 individuals?
- Six did not present symptoms recognized as MPXV infection => What did the seventh present with? (and if he had symptoms recognized as MPXV infection, why were they included in the study?)

Main text:

- Although the authors provided some more clarity about the recruitment of participants and the study design in their answers to my comments, these aspects are still not entirely clear from the current flow of the manuscript. I understand that Nature Communications requires to mention the methods section at the end of the study, but essential information about the way participants are recruited and examined/surveyed should be mentioned early on for correct interpretation of the results. Currently, for example, it is not clear how symptom status was assessed at the time of sampling and, in line with this, how potential study participants were included or excluded from the study. How was the flow of patients? How were they recruited? Which information were they given prior to study participation? What happened to those who did report symptoms? If this study was planned prospectively and the initial aim was to assess the prevalence of asymptomatic mpox, then why was symptom status not assessed at the time of inclusion? As such, the inclusion criteria are not clear and I do not agree with the statement that "no participants were excluded from the study".

"While this may introduce some bias, we did not observe significant differences in the number of sexual partners in the last 30 days or the history of exchanging sex for money, gifts, or favors between participants with positive and negative MPXV infection results. " => I do not agree with this argument: a comparison between mpox positive and negative patients does not provide information about the generalizeability of results.

REVIEWER COMMENTS

Reviewer #1 (Remarks to the Author):

Comment: Authors have addressed my comments and have modified the objective of the study by clearly mentioning that they included asymptomatic and participants with mild/unrecognized symptoms.

However, my comment regarding the discussion section - i.e. the discussion section is too long and mixes scientific virological findings with socio-behavioral assumptions and on the acceptability/feasibility on self-sampling, that are not supported by the results presented herein and/or were not part of the objectives of the present study - has not been addressed as authors state that they disagree with my comment.

Answer: We appreciate your comment and concerns regarding the discussion section of our paper. We acknowledge that the discussion section may have been lengthy and that it includes a combination of scientific virological findings and socio-behavioral assumptions related to the acceptability and feasibility of self-sampling. We have taken this into consideration and we revised the discussion accordingly, ensuring that the new version is more concise and grounded in empirical evidence.

To address the concern regarding unsupported statements, we removed those parts that they were not adequately substantiated by existing literature. Furthermore, we understand the reviewer's suggestion to tone down claims related to transmission, benefits of the study, and differences between groups included in the study, especially considering the small sample size. Accordingly, we have deleted these sections from the text or modified their tone.

We appreciate your valuable feedback, and we trust that we have successfully incorporated the required revisions to enhance the quality of the discussion section.

Reviewer #2 (Remarks to the Author):

This is a very interesting paper that focuses on asymptomatic infections of Mpox. It would be interesting to understand possible further transmission although I understand this may not be within the scope of the paper. I have some questions and minor amendments to the manuscript below:

Methods

Comment: More details on how the authors arrive at a sample size of 113 participants? How many were excluded and the dropout rate. What biases did this introduce.

Answer: Thank you very much for your feedback. We managed to recruit a sample of 113 participants through intermittent dissemination campaigns on Instagram, Facebook, WhatsApp and the community centre website. The campaigns indicated the

possibility of get tested for MPXV through a self-sampling intervention free of charge at the headquarters of the collaborating community centre. Participants with eligible criteria were invited to attend the collaborating community centre to get tested for MPXV. No participant was excluded in the study. More detailed information on the content of the dissemination campaigns has been included in the Methods section. Regarding the potential introduced biases, see the answer to the third comment.

Comment 2: Thank you for including this context in the paper.

Answer to comment 2: Thank you for your valuable feedback and positive acknowledgement. We greatly appreciate your recognition of the context included in our paper.

Comment: Was sample size a consideration in study design?

Answer: Yes, It was. It was expected to include 150 participants in the study. A random sample of 142 individuals was established as sufficient to estimate, with 95% confidence and an accuracy of +/- 2 percentage units, a population percentage that was expected to be around 1.5% (De Baetselier I, Van Dijck C, Kenyon C, Coppens J, Michiels J, de Block T, Smet H, Coppens S, Vanroye F, Bugert JJ, Giral P, Zange S, Liesenborghs L, Brosius I, van Griensven J, Selhorst P, Florence E, Van den Bossche D, Ariën KK, Rezende AM, Vercauteren K, Van Esbroeck M; ITM Monkeypox study group. Retrospective detection of asymptomatic monkeypox virus infections among male sexual health clinic attendees in Belgium. *Nat Med.* 2022 Nov;28(11):2288-2292. doi: 10.1038/s41591-022-02004-w. Epub 2022 Aug 12. PMID: 35961373; PMCID: PMC9671802.). Finally, 113 individuals participated in the study.

Comment 2: How did you establish this random sample was sufficient? I would include further context.

Answer to comment 2: Thank you for your additional question. To establish the adequacy of the random sample of 142 individuals, a sample size calculation was conducted based on statistical considerations. The aim was to estimate, with 95% confidence and an accuracy of +/- 2 percentage units, the population percentage expected to be approximately 1.5%. The expected prevalence was based in a previous study performed in Belgium among male sexual health clinic attendees with no symptoms of Mpox (De Baetselier I, Van Dijck C, Kenyon C, Coppens J, Michiels J, de Block T, Smet H, Coppens S, Vanroye F, Bugert JJ, Giral P, Zange S, Liesenborghs L, Brosius I, van Griensven J, Selhorst P, Florence E, Van den Bossche D, Ariën KK, Rezende AM, Vercauteren K, Van Esbroeck M; ITM Monkeypox study group. Retrospective detection of asymptomatic monkeypox virus infections among male sexual health clinic attendees in Belgium. *Nat Med.* 2022 Nov;28(11):2288-2292. doi: 10.1038/s41591-022-02004-w.

Epub 2022 Aug 12. PMID: 35961373; PMCID: PMC9671802). This calculation relied on standard statistical techniques that take into account the expected population size, anticipated variability in the data, and the desired levels of confidence and precision. Based on these considerations, it was determined that a sample of 142 individuals would be sufficient to achieve the study's objectives. However, it is important to acknowledge that scientific studies are subject to limitations and assumptions, and the optimal sample size may vary depending on various factors such as the nature of the phenomenon under study and the characteristics of the target population. In this specific case, although the initial plan was to include 150 participants in the study, 113 individuals ultimately participated due to logistical or other circumstances.

Despite a decrease in the precision of the estimates resulting from a smaller sample size than initially expected (113 participants instead of the anticipated 150), the confidence intervals have remained significant but wider because the obtained prevalence was higher compared to the prevalence used for determining the sample size.

In the revised version of the manuscript, we have incorporated a concise description of the preceding study that was utilized to establish the anticipated prevalence. This addition aims to offer further context, thereby addressing your request for additional clarification.

Methods section:

“After the start of vaccination and the decrease in new cases of mpox in Spain, the interest of the target population in a screening intervention dropped considerably and it was impossible to reach the expected n (150).”

Comment 2: I would not use the word “impossible” here but rather difficult.

Answer to comment 2: We appreciate this comment and we have corrected the text following the recommendation of the reviewer.

We have also discussed how this affects study aim/outcome:

Study limitations (Discussion section): Although the precision of the estimates decreased due to a reduction in the expected number of participants, the confidence intervals have remained significant but wider because the obtained prevalence was higher than the prevalence used to calculate the sample size.

Comment: What biases were present during recruitment?

Answer: The authors acknowledge the reviewer's concern regarding the potential biases present during the recruitment. The collaborating center work mainly with gay, bisexual and other men who have sex with men and trans women. They have a specific program for sex workers and most of them are migrant. This could explain why among the studied

population there was a great percentage of migrants and sex workers. This could be a bias and the first limitation reported by the authors in the article is: "The study population is not representative of GBMSM and TW in Catalonia as we used an opportunistic sample". The following text has been added: "However, it is worth noting that the sample had a notably high proportion of migrants and sex workers, likely due to the involvement of a collaborating center with a specific program for this population. While this may introduce some bias, we did not observe significant differences in the number of sexual partners in the last 30 days or the history of exchanging sex for money, gifts, or favors between participants with positive and negative MPXV infection results".

Comment 2: Is the sample size adequate to say "we did not observe significant differences in the number of sexual partners in the last 30 days or the history of exchanging sex for money, gifts, or favors between participants with positive and negative MPXV infection results" with "significant" here implied as a statistical term.

Answer to comment 2: In response to Comment 2, we acknowledge that the term 'significant' was used to indicate a statistical significance. The comparison between participants with positive and negative MPXV infection results regarding the number of sexual partners in the last 30 days and their history of exchanging sex for money, gifts, or favors was assessed using a Chi-square analysis. We have included the data mentioned in the paragraph to accurately reflect the scope and limitations of our study. We appreciate your attention to this matter and hope this clarification addresses your concern.

Comment: "participants had received the Mpox vaccine in their childhood or in the previous 12 months, respectively" - do the authors mean the smallpox vaccine in childhood here?

Answer: Yes, the authors referred to the smallpox vaccine. It has been clarified in the new version of the manuscript: "History of small pox and Mpox vaccination".

Comment 2: Did any participants receive the mpox vaccine or was it all smallpox?

Answer to comment 2: Thank you for your comments and for bringing up this important point. Upon further review, we have realized that there was an error in the previous revised version of the manuscript. We apologize for the confusion caused. In fact, all participants received the smallpox vaccine, as there is currently no specific vaccine available for Mpox. We appreciate your attention to detail, and we have made the necessary corrections in the updated version of the manuscript to accurately reflect that all participants received the smallpox vaccine.

Comment: "38 (33.63%) had had sex in exchange for money, gifts or favours" - this suggests a recruitment bias that would be relevant to discuss.

Answer: The authors acknowledge the potential for recruitment bias related to the proportion of participants who reported exchanging sex for money, gifts, or favors. However, it is important to note that this high percentage may reflect the population served by the collaborating center that recruited the study participants, which had a specific program for sex workers and migrants. We have added a discussion of this potential bias in the revised manuscript (See Limitations of the study) to provide greater transparency and contextualization for the findings.

Comment 2: Thank you for including this context.

Answer to comment 2: We sincerely appreciate your valuable contribution to improving the accuracy and comprehensiveness of our research.

Comment: Some high CT values "36.79" that would perhaps warrant a retest per CDC advice "If you obtain a high Ct value (generally ~34 or higher), CDC recommends to immediately re-extract and re-test to ensure there was no cross-contamination."(https://www.cdc.gov/locs/2022/08-23-2022-Lab-Advisory-Monkeypox_Virus_Testing_Considerations_Prevent_False_Positive_Test_Results.html) . Worth including this in the limitations.

Answer: We have reviewed all the raw data and PCR curves in all samples including those with high Ct values that are also recommended by the CDC. In samples with high Ct re-extract and re-test was not performed due to the small volume of original sample to carry out all the experiments.

We have included the following text in the Study limitations (Discussion Section):

“Fourthly, due to the high sensitivity of PCR techniques, there is the possibility of obtaining false positive results. In this situation, Ct values are high (Ct>34) and are associated with low viral loads and the laboratory should re-extract and re-amplify the original sample or a new sample requested if this is not possible²⁹. In our study, there were three samples with high Cts (64 pharynx; 72 anal; 81 pharynx). Re-testing these samples was not possible due to the low volume left. However, the PCR performed after the viral culture for MPXV was positive for one sample (sample 72). Previous observations have shown that pharyngeal and anal samples exhibit higher Cts in comparison to samples obtained from lesions^{4,7}. Furthermore, good practices were followed to prevent contamination and erroneous diagnoses, and negative controls produced the expected result. Moreover, the study's conclusions rely on samples in which the ability of MPVX to replicate has been verified and validated using diverse and independent methods.”

Comment 2: Thank you for including this context.

Answer to comment 2: Thank you very much for your feedback, it has allowed us to address this oversight and provide additional context in the revised version.

Comment: Line 422 "MPXV infection prevalence was estimated by calculating the proportion of individuals with a positive result over the total of individuals with a returned and valid sample. Confidence interval of 95% was calculated." - This needs expanding. Are they binomial confidence intervals? If so what method?

Answer: Thank you for your comment and for bringing attention to the need for further clarification regarding the calculation of confidence intervals for MPXV infection prevalence. We used the binomial method to estimate 95% confidence intervals for the proportion of individuals with positive test results. We have included this information in the revised manuscript to provide more transparency and to aid the reproducibility of our study.

Comment 2: Thank you for including this. I would still include the method used to calculate the binomial uncertainty.

Answer to comment 2:

We have taken into account your request to include the method used to calculate the binomial uncertainty in the manuscript. In the revised version of the article, we have added a reference* that specifically details how this calculation was performed. We appreciate your constructive feedback and hope that this update meets your expectations

*Agresti, A. & Coull, B. A. Approximate is Better than 'Exact' for Interval Estimation of Binomial Proportions. Am. Stat. 52, 119–126 (1998).

Please find below the used formula:

$$P \pm z_{1-\alpha/2} \sqrt{\frac{p(1-p)}{n}}$$

Comment: A suitable alternative method that authors may want to consider for investigating the characteristics of the patients with mpox could be logistic regression.

Answer: Thank you for your comment and suggestion regarding the use of logistic regression for investigating the characteristics of patients with mpox. We appreciate your input and agree that logistic regression could be a useful alternative method for this analysis. However, given the relatively small number of positive cases (only seven individuals), we ultimately concluded that the sample size was insufficient to fit a model, particularly for covariates with more than two categories or those with unbalanced

proportions. Furthermore, following the recommendation of other reviewer we removed all the statistical analyses that were clearly not driven by a prior hypothesis.

Comment 2: *I think this is a fair justification.*

Answer to comment 2: We really appreciate your input and the opportunity to provide further clarification.

Reviewer #3 (Remarks to the Author):

Comment: *The manuscript title is not grammatically correct.*

Answer: We appreciate your comment and we have corrected the title of the new version of the manuscript: A SELF-SAMPLING SCREENING STRATEGY ADDRESSED TO ASYMPTOMATIC OR UNRECOGNIZED MONKEYPOX VIRUS INFECTION IN GAY, BISEXUAL, AND OTHER MEN WHO HAVE SEX WITH MEN AND TRANS WOMEN IN SPAIN

Likewise for some of the revisions in the text.

Abstract:

Comment: - *I suggest to simplify the first sentence as "we aimed to assess the prevalence of monkeypox virus (MPXV) among gay, bisexual, and other men who have sex with men and trans women (TW) without symptoms or with unrecognized Mpox symptoms, using a self-sampling strategy"*

Answer: Thanks for your suggestion, we have simplified the first sentence of the abstract following your recommendation.

Comment: - *113 individuals participated: 89 men + 17 TW = 106 => what was the gender of the remaining 7 individuals?*

Answer: We appreciate the reviewer for their comment. In our study, a total of 113 individuals participated. Out of these, 89 were identified as male, 17 as transgender women, and 3 participants identified as non-binary gender. Additionally, 4 participants chose not to disclose their gender or did not provide a clear response. This information has now been clarified in the article. We appreciate your attention and comments on our work.

Comment: - *Six did not present symptoms recognized as MPXV infection => What did the seventh present with? (and if he had symptoms recognized as MPXV infection, why were they included in the study?)*

Answer: We appreciate the reviewer's comment. In our study, we had information about symptomatology of 6 out of 7 Mpox positive cases. Unfortunately, specific

information regarding the seventh positive case and their symptoms is not available to us. Due to the unavailability of data on the seventh case, we were unable to include their symptoms in the study. We acknowledge the limitation in not providing details on this particular case. We have clarified this issue in the revised version of the manuscript as follows: "Regarding presentation of symptoms, there was no information available regarding one of the participants with a positive MPXV result (1/7), two (2/7) positive-testing participants reported having no symptoms before testing, or 21 days after. One (1/7) had no symptoms before testing and reported having fever, exhaustion, sore throat and a skin lesion in the 21 days following testing positive. Three (3/7) participants reported the following symptoms before testing: a swollen inguinal lymph node, fever, exhaustion and a skin lesion and none of those participants connected these symptoms with MPXV infection. (Table 3)." We thank the reviewer for bringing this to our attention and for their valuable input

Main text:

Comment: - Although the authors provided some more clarity about the recruitment of participants and the study design in their answers to my comments, these aspects are still not entirely clear from the current flow of the manuscript. I understand that Nature Communications requires to mention the methods section at the end of the study, but essential information about the way participants are recruited and examined/surveyed should be mentioned early on for correct interpretation of the results. Currently, for example, it is not clear how symptom status was assessed at the time of sampling and, in line with this, how potential study participants were included or excluded from the study. How was the flow of patients? How were they recruited? Which information were they given prior to study participation? What happened to those who did report symptoms? If this study was planned prospectively and the initial aim was to assess the prevalence of asymptomatic mpox, then why was symptom status not assessed at the time of inclusion? As such, the inclusion criteria are not clear and I do not agree with the statement that "no participants were excluded from the study".

Answer: We appreciate the valuable comments provided by the reviewer. With regard to the concern about the assessment of symptom status during sampling and the inclusion or exclusion of potential study participants, we would like to clarify that clinical evaluation of symptoms did not take place during the recruitment process. However, participants were explicitly informed that they would be ineligible to participate if they exhibited any symptoms compatible with Mpox. This precautionary measure was implemented to safeguard against the inclusion of individuals with potential Mpox symptoms in the study population. It has been clarified in the new version of the manuscript.

We appreciate the reviewer's comment regarding the flow of patients, recruitment methods, and information provided to participants prior to study participation. The study utilized various platforms, including Instagram, Facebook, Whatsapp, and the community center website, to disseminate information through intermittent campaigns. These campaigns aimed to inform individuals about the opportunity to undergo free self-sampling intervention for MPXV at the collaborating community center. Participants who met the eligible criteria were invited to visit the community center for MPXV testing. Importantly, no participants were excluded from the study. During the recruitment process, the community center staff checked if potential participants met the inclusion criteria. Brief explanations about the project were provided, and signed informed consent was obtained on paper. It is worth noting that clinical evaluation of symptoms was not conducted at the time of recruitment. However, participants were explicitly informed that they could not participate in the study if they presented any symptoms compatible with Mpox. Following the recommendations of the reviewer, the recruitment process has been clarified and an explanation of the information provided to participants has been included in the new version of the manuscript.

With regard to the question, "What happened to those who did report symptoms?", it is important to note that none of the participants reported any symptoms during the recruitment phase. This information has been clarified and included in the revised version of the manuscript.

In response to the question, "If this study was planned prospectively and the initial aim was to assess the prevalence of asymptomatic Mpox, then why was symptom status not assessed at the time of inclusion?", we would like to clarify that our study was designed as a cross-sectional prospective non-randomized study rather than a purely prospective study. We would like to clarify that our study is a cross-sectional study that includes a question regarding the onset of symptoms within 21 days following the test. However, it is important to note that the inclusion of this question does not imply that our study is prospective in nature. We apologize for any confusion caused and appreciate the opportunity to address this concern. While we acknowledge that no clinical assessment of symptoms was conducted at the time of inclusion, it is important to note that participants were explicitly informed that they would be ineligible to participate if they exhibited any symptoms compatible with Mpox. This precautionary measure was implemented to ensure that individuals with potential Mpox symptoms were not included in the study population.

In response to the concern raised about the clarity of the inclusion criteria and the statement that "no participants were excluded from the study," we appreciate the reviewer for bringing this to our attention. We agree that the inclusion criteria were not clearly outlined in the initial version of the manuscript. We have taken this feedback into

consideration and have made necessary revisions to provide a more explicit description of the inclusion criteria in the updated version of the manuscript. Thank you for your valuable input, and we apologize for any confusion caused by the lack of clarity in the initial presentation of the inclusion criteria.

Comment: "While this may introduce some bias, we did not observe significant differences in the number of sexual partners in the last 30 days or the history of exchanging sex for money, gifts, or favors between participants with positive and negative MPXV infection results. " => I do not agree with this argument: a comparison between mpox positive and negative patients does not provide information about the generalizability of results.

Answer: Thank you for your input, and we appreciate the opportunity to clarify our argument. We recognize that the study population is not representative of GBMSM and TW in Catalonia, as highlighted in the study's limitations. The use of an opportunistic sample introduces selection bias and limits the generalizability of our findings. While we acknowledge the limitations, such as the potential bias from the high proportion of migrants and sex workers, the comparison between participants with positive and negative MPXV infection results served as an illustrative example. Our aim was to demonstrate that despite these biases, we did not observe significant differences in specific factors between MPXV positive and negative individuals. The intention was not to generalize the results but rather to highlight the lack of differences in those particular variables.

General answer to all reviewers:

Thank you for your feedback and for highlighting the need for further attention to address the concerns raised by our reviewers in the manuscript. We appreciate your guidance and constructive criticism. In response to your comments, we would like to assure you that we are fully committed to addressing the raised concerns in a comprehensive and appropriate manner. We dedicated additional effort to thoroughly revise the manuscript, taking into account the reviewers' comments and suggestions. We value your input and understand the importance of meeting the expectations of both the reviewers and the scientific community. Once again, we express our gratitude for your valuable insights and we trust we have made the necessary revisions to address the raised concerns effectively.